# Multiscale effects of excitatory-inhibitory homeostasis in lesioned cortical networks: A computational study

**Francisco Páscoa dos Santos**[1,2]*, **Jakub Vohryzek**[3,4], **Paul F. M. J. Verschure**[5]

**1** Eodyne Systems SL, Barcelona, Spain, **2** Department of Information and Communication Technologies, Universitat Pompeu Fabra (UPF), Barcelona, Spain, **3** Centre for Brain and Cognition, Computational Neuroscience Group, Department of Information and Communication Technologies, Universitat Pompeu Fabra, Barcelona, Spain, **4** Centre for Eudaimonia and Human Flourishing, Linacre College, University of Oxford, United Kingdom, **5** Donders Institute for Brain, Cognition and Behavior, Radboud University, Nijmegen, The Netherlands

* f.pascoadossantos@gmail.com

**Data Availability Statement:** The code necessary to run the model developed for this work and to perform all the relevant analysis is available at:

## Abstract

Stroke-related disruptions in functional connectivity (FC) often spread beyond lesioned areas and, given the localized nature of lesions, it is unclear how the recovery of FC is orchestrated on a global scale. Since recovery is accompanied by long-term changes in excitability, we propose excitatory-inhibitory (E-I) homeostasis as a driving mechanism. We present a large-scale model of the neocortex, with synaptic scaling of local inhibition, showing how E-I homeostasis can drive the post-lesion restoration of FC and linking it to changes in excitability. We show that functional networks could reorganize to recover disrupted modularity and small-worldness, but not network dynamics, suggesting the need to consider forms of plasticity beyond synaptic scaling of inhibition. On average, we observed widespread increases in excitability, with the emergence of complex lesion-dependent patterns related to biomarkers of relevant side effects of stroke, such as epilepsy, depression and chronic pain. In summary, our results show that the effects of E-I homeostasis extend beyond local E-I balance, driving the restoration of global properties of FC, and relating to post-stroke symptomatology. Therefore, we suggest the framework of E-I homeostasis as a relevant theoretical foundation for the study of stroke recovery and for understanding the emergence of meaningful features of FC from local dynamics.

## Author summary

Excitatory-inhibitory (E-I) balance is an essential feature of cortical network function and is known to be maintained locally by homeostatic plasticity. In this work, we explore how the effects of such balancing mechanisms extend beyond the mesoscale and contribute to the maintenance of relevant macroscale properties of functional connectivity. More specifically, we suggest local E-I homeostasis is tied to the reorganization of large-scale functional networks following a focal lesion, providing an explanation for the recovery of relevant functional properties at a global level. To that end, we built a network model of

https://gitlab.com/francpsantos/stroke-e-i-homeostasis.

**Funding:** FPS is supported by the European Commission through the euSNN project (MSCA-ITN ETN H2020—ID 860563). JV is supported by EU H2020 FET Proactive project Neurotwin (ID no. 101017716). PFMJV is supported by Virtual Brain Cloud (H2020 ID 826421), AISN (HE, 101057655), RGS@HOME (EIT Health—ID 19277) and PHRASE (EIC, 101058240). The funders had no role in study design, data collection and analysis, decision to publish, or preparation of the manuscript.

**Competing interests:** I have read the journal's policy and the authors of this manuscript have the following competing interests: FPS is employed by the company Eodyne Systems SL. PFMJV is founder and shareholder of Eodyne Systems S.L., which aims at bringing scientifically validated neurorehabilitation and education technology to society

interacting neural masses, constrained by the human connectome and accounting for local homeostasis of E-I balance. We show that this mechanism drives the recovery of properties such as modularity and small-worldness after simulated lesions and that the resultant patterns of change in excitability can be related to known late-onset symptoms of stroke such as seizures, depression, and chronic pain, in a lesion-dependent manner. Therefore, we propose E-I homeostasis as a relevant driver of recovery in lesioned networks and a contributing factor to the etiology of specific side effects of stroke, emerging as a byproduct of lesion-dependent changes in local excitability.

## 1. Introduction

Stroke, characterized by neural tissue necrosis (i.e. lesion) due to oxygen loss after occlusion or hemorrhage of vessels supplying blood to the brain, is one of the leading causes of disability, with a significant negative impact on patient life quality [1] due to its debilitating symptoms, ranging from motor deficits to impaired higher-order functions such as attention and memory [1,2]. Besides these symptoms, stroke patients tend to develop long-term side effects such as seizures (in some cases evolving into epilepsy) [3–5], chronic pain [6,7], depression [8–10] and chronic fatigue [11]. Cognitive deficits can occur in both a lesion-specific fashion, e.g. attention, memory, executive control, and language, and be manifested as a diffuse neuronal dysfunction, with a more uniform profile of mental slowing and memory and executive deficits [12]. This heterogeneity in symptoms and side effects raises the need to better understand the mechanisms through which these symptoms emerge, to better predict their occurrence and to inform therapeutical approaches. This task is made difficult not only by the heterogeneity in lesions, but also since their consequences on neural activity and connectivity often spread beyond lesioned areas [13,14]. This phenomenon, first described by Konstantin von Monakow in 1914 [15], is known as diaschisis. Although its initial conception pertained to acute changes in the excitability of regions distant from the lesion, today the concept has been expanded to include global changes in connectivity [14]. This might include a range of deficits in functional connectivity (FC), from disconnection between particular areas [16–19] to structural-functional decorrelation [20]. However, it is considered that the most robust disruptions, found to correlate with changes in function, are decreased homotopic interhemispheric functional connectivity and increased functional connectivity between regions that were not previously connected [21], manifesting through a loss of modularity [22,23]. Modularity, a property of networks that have strong connectivity within node communities, with sparser connections between them, has been observed in human functional and structural networks and is considered to reflect an appropriate balance between segregation and integration of networks, underlying functional specialization [24,25]. Importantly, modularity is significantly disrupted following a stroke and is recovered in the following months, with the magnitude of recovery correlating with improvement in higher-order functions such as attention and working memory [22]. Similarly, small-worldness, a property of networks where most nodes are not neighbors, but can be reached through a short path through highly connected nodes (hubs) [26], is lost after a stroke and subsequently recovered [22]. Besides affecting structural and functional connectivity, stroke lesions may have comparable effects on cortical network dynamics. Modeling studies suggest significant post-lesion effects on dynamical features such as metastability, quantifying the ability of a network to flexibly switch between synchronous and asynchronous states [27] or criticality, a property of brain networks underlying balanced propagation of activity [28]. In addition, recent results suggest that FC dynamics, which have

been related to cognitive ability [29,30], are affected in stroke patients, with impairment-dependent differences between subjects [31], and that specific altered patterns of FC dynamics can been tied to functional recovery [32]. Furthermore, recent results suggest that severely impaired patients are associated with increased occurrence of transient states associated with common biomarkers of stroke, such as decreased interhemispheric homotopic connectivity and modularity [33] Therefore, the post-stroke loss, and subsequent recovery, of global properties of FC and network dynamics, raise the question of how the human cortex coordinates the restoration of such properties on a large scale.

Several studies have reported persistent long-term increases in excitability in the period following stroke, both in rodent models of the disease [34–36] and in human patients [37–39]. Such increases have been related to several factors, from increased glutamatergic receptor density [40], prolonged excitatory postsynaptic potentials [34] or, more importantly, decreased GABAergic signaling [36,41–43]. Indeed, studies in stroke patients indicate that not only is there a longitudinal decrease in the availability of GABAergic neurotransmitters in the cortex [38], but that its magnitude correlates with behavioral recovery [39]. Therefore, as previously suggested [13,44,45], it is likely that these long-term changes play a significant role in stroke recovery and might result from mechanisms that maintain excitatory-inhibitory (E-I) balance in cortical networks, following an acute decrease in the levels of incoming excitation to areas that share white-matter tracts with the lesioned region.

Indeed, research supports E-I balance as a pivotal feature of cortical networks [46–49], which maintain a close-knit balance between the levels of excitation and inhibition arriving at individual pyramidal neurons [50–52]. In addition, criticality, an emergent signature of E-I balance, has been consistently observed in neural dynamics [53–56] and is relevant for the optimization of functions ranging from high dynamic ranges to information capacity and transmission [57–59]. Given its relevance to neural function, cortical neurons have mechanisms of homeostasis that maintain E-I balance [60], from synaptic scaling of excitatory synapses to regulation of intrinsic excitability [61–65]. Of particular interest is the scaling of incoming inhibitory synapses by pyramidal neurons, which has been shown to occur after perturbations such as sensory deprivation [64] and to be a strong factor underlying sensory co-tuning, memory stability [48] and criticality in cortical networks [66]. Importantly, these processes work on long timescales of hours to days in mice [60] and up to several weeks in monkeys, depending on the type of perturbation [67]. Therefore, it is likely that such homeostatic mechanisms might participate in stroke recovery [13,44,45] and underlie the long-term changes in excitability observed in patients [38,39]. In addition, it could be possible, as previously suggested [68], that homeostatic plasticity mechanisms are not only responsible for restoring local E-I balance but also contribute to recalibrating global properties of FC. Therefore, E-I homeostasis could potentially explain the long-term local changes in excitability and the recovery of global dynamics and FC properties simultaneously.

On this subject, not only have previous modeling studies shown the importance of E-I homeostasis to accurately reproduce cortical dynamics [69] and functional connectivity [70–72], but also that it might be involved in stroke recovery. The study of Vattikonda and colleagues [68] showed that the restoration of E-I balance, through inhibitory synaptic scaling, further contribute to the recovery of FC in a lesion-dependent manner. In addition, models fitted to FC from stroke patients showed reduced local inhibition compared to healthy controls [73]. Such approaches, however, lack a detailed exploration of what E-I homeostasis entails regarding the changes in excitability that are driving this adaptive process how they are distributed across the brain. This understanding is relevant not only to better link the action of E-I homeostasis to current knowledge on post-stroke changes in excitability [36,38,39] but also to elucidate the etiology of stroke symptomatology, such as post-stroke seizures [3], depression

[10] and chronic pain [7], which have been tied to changes in excitability. E-I homeostasis could then explain why stroke patients display an increased propensity to develop such symptoms, framing them as side-effects of homeostatic plasticity attempting to restore local E-I balance.

Therefore, we hypothesize that E-I homeostasis not only plays an important role in the maintenance of E-I balance at the mesoscale but also in the recovery of macroscale properties of FC (i.e. modularity and small-worldness). In this modeling study, we aim to explore the involvement of E-I homeostasis in recovery from localized lesion in large-scale networks of interacting nodes and the subsequent changes in excitability it entails, probing into the multiscale impact of the regulation of E-I balance in local and global network features of cortical networks. To that end, we simulate gray-matter lesions in a network model constrained by the structural connectome of the human cortex, including local E-I homeostasis mechanisms. Our main goal is then to study the long-term mesoscale changes in excitability observed in lesioned brain networks through the lens of homeostatic plasticity, tying them to the macroscale recovery of FC and suggesting a novel process participating in the etiology of late-onset side effects of stroke previously related to altered cortical excitability, such as epilepsy, depression and chronic pain. In addition, considering the specific case of stroke, will help elucidate generic features of multiscale cortical organization and the optimization objectives it serves. More generally, we aim to study how large-scale emergent properties of FC, such as modularity, depend on ruling principles of mesoscale dynamics, thus advancing the discussion on the governing principles of emergence.

## 2. Methods

### 2.1. Empirical data

**2.1.1. Structural connectivity.**   In order to derive structural connectivity matrices of 78x78 dimensions, we used a probabilistic tractography-based normative connectome from the leadDBS toolbox (https://www.lead-dbs.org/). This normative connectome comes from 32 healthy participants (mean age 31.5 years old ± 8.6, 14 females) generated as part of the Human Connectome Project (HCP - https://www.humanconnectome.org) from diffusion-weighted and T2-weighted Magnetic Resonance Imaging data recorded for 89 minutes on a specially set up MRI scanner with more powerful gradients to the standard models. The HCP data acquisition details can be found in the Image & Data Archive (https://ida.loni.usc.edu/). For the diffusion imaging, DSI studio (http://dsi-studio.labsolver.org) with a generalized q-sampling imaging algorithm was used. Furthermore, a white-matter mask, derived from the segmentation of the T2-weighted anatomical images was applied to co-register the images to the b0 image of the diffusion data using the SPM 12 toolbox (https://www.fil.ion.ucl.ac.uk/spm/software/spm12/). Then, each participant was sampled with 200 000 most probable tracts. The tracts were transformed to the standard space (MNI space) by applying a nonlinear deformation field, derived from the T2-weighted images via a diffeomorphic registration algorithm [74]. The individual tractograms were then aggregated into a joint dataset in MNI standard space resulting in a normative tractogram representative of a healthy young adult population and made available in the leadDBS toolbox [75]. Finally, to obtain structural connectomes from the normative connectome in our desired parcellation–the Anatomic Automatic Labeling (AAL) atlas [76] -, we calculated the mean tracts between the voxels belonging to each pair of brain regions. Tract lengths between two given brain regions, in turn, were extracted by computing the three-dimensional distance across all the voxels that the fiber passes through, similarly to the method used in [77].

**2.1.2. BOLD fMRI time series.** The data from healthy controls used to fit the model were obtained from the public database of the Human Connectome Project (HCP), WU-Minn Consortium (Principal Investigators: David Van Essen and Kamil Ugurbil; 1U54MH091657) funded by the 16 NIH Institutes and Centers that support the NIH Blueprint for Neuroscience Research; and by the McDonnell Center for Systems Neuroscience at Washington University. [78].

The specific data used in this project was obtained from 100 unrelated subjects from the HCP database (mean age 29.5 years old, 55% females). Each subject underwent four resting-state fMRI sessions of about 14.5 minutes on a 3-T connectome Skyra scanner (Siemens) with the following parameters: TR = 0.72 s, echo time = 33.1 ms, field of view = 208x180mm, flip angle = 52˚, multiband factor = 8, echo time = 33.1 with 2x2x2 isotropic voxels with 72 slices and alternated LR/RL phase encoding. For further details on the data acquisition and standard processing pipeline, please consult [79] and https://www.humanconnectome.org/study/hcp-young-adult/data-releases. In this work, we used the data from the first session of the first day of scanning.

The AAL atlas was further used to parcellate the voxel-based data into 90 anatomically distinct cortical and subcortical regions, excluding the cerebellum. For this work, we then exclude the 12 subcortical regions, given that our modeling approach is focused on cortical dynamics (see section 2.2). Therefore, after averaging BOLD signals associated with each of the 78 cortical regions, data was reduced to size 78 areas X 1200 TR.

## 2.2. Neural mass model

To model the activity of individual cortical regions we make use of the Wilson-Cowan model of coupled excitatory and inhibitory populations [70,80] (Fig 1A). As a mean-field approach, the Wilson-Cowan model is based on the assumption that the neural activity of a determined population of neurons can be described by its mean at a given instant in time [81]. Shortly, the equations describing the firing-rate dynamics of $N = 78$ coupled excitatory ($r^E$) and inhibitory ($r^I$) populations, adapted from [70], can be written as:

$$\tau_E \frac{dr_i^E(t)}{dt} = -r_i^E(t) + F\left[c_{EE}r_i^E(t) - c_{EI,i}(t)r_i^I(t) + C\sum_{j=1}^{N} W_{ij}r_j^E(t-\tau_{ij}) + \xi(t) + P\right],$$

$$\tau_I \frac{dr_i^I(t)}{dt} = -r_i^I(t) + F\left[c_{IE}r_i^E(t) + \xi(t)\right] \tag{1}$$

where $c_{EE}$ represents the recurrent coupling in excitatory populations, $c_{IE}$ represents the coupling from excitatory to inhibitory populations and $c_{EI,i}$ represents the local coupling from inhibitory population $i$ to excitatory population $i$. $C$ is a scaling factor for structural connectivity, formally called global coupling, and $\xi$ is additive Gaussian noise with variance 0.01. $W_{ij}$ represents the structural connection between nodes $i$ and $j$ in the large-scale network and is constrained by human structural connectivity data (see section 2.1.1). $\tau_{ij}$, in turn, represents the conduction delay between regions $i$ and $j$ and is determined according to empirical white-matter tract length, by dividing tract lengths by a given conduction speed. Long-range connections are only implemented between excitatory neural masses, given the evidence that long-range white matter projections are nearly exclusively excitatory [82], and following the state-of-the-art in large-scale modeling [69,70,72,83]. In addition, similarly to [70], we include a term $P$ that can be understood as the intrinsic excitability of neural masses. $F(x)$ is a sigmoid function representing the activation function of a population of neurons, determining its

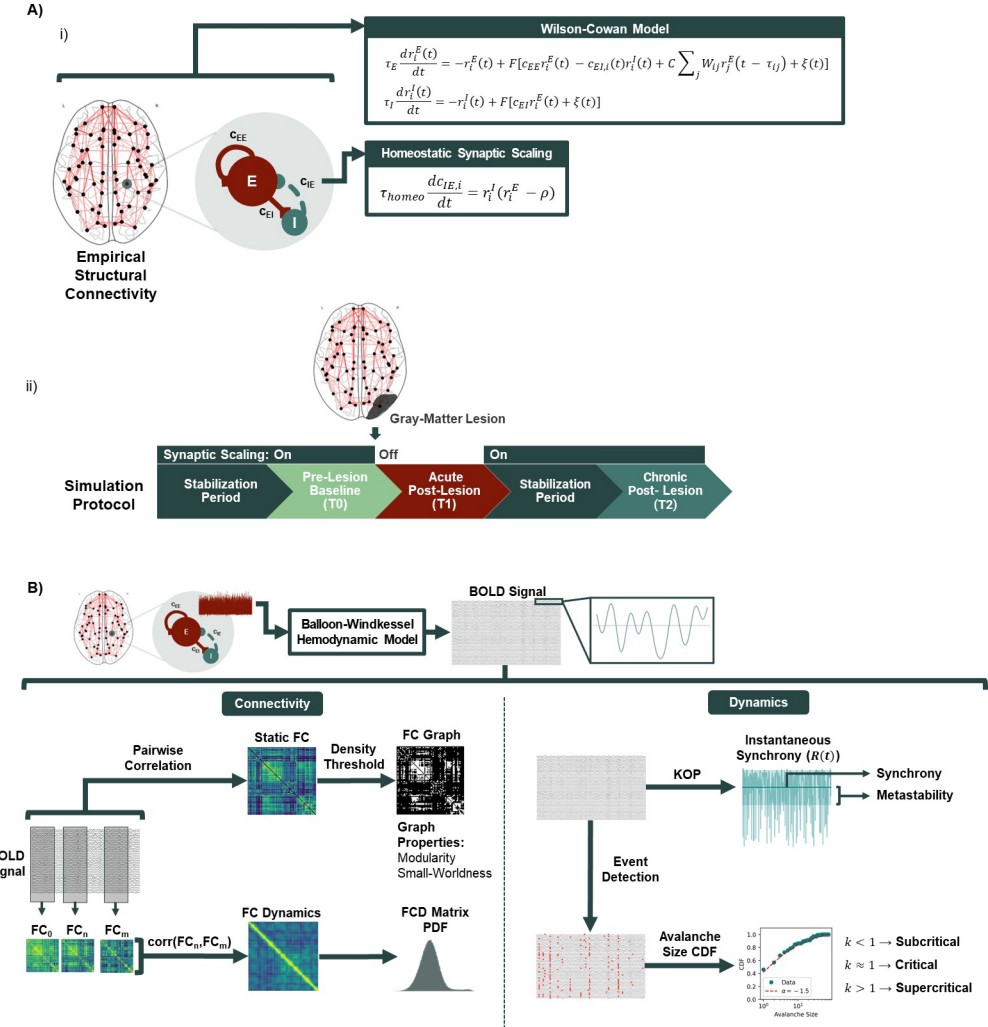

**Fig 1. Computational model and data analysis.** A) Model Architecture and simulation protocol. i) Cortical dynamics were modeled using a system of neural masses connected through long-range excitatory connections derived from DTI from healthy subjects. Local activity was simulated using the Wilson-Cowan model of coupled excitatory and inhibitory populations, with the addition of homeostatic plasticity regulating inhibitory synapses, with the goal of maintaining excitatory firing rates at a target level ($\rho$). ii) To study the effects of stroke on functional connectivity, the model is first run until a steady state is reached in terms of local inhibitory weights (T0), after which a lesion is applied by removing all connections from the lesioned area in the structural connectivity matrix. Acute activity is then extracted before plasticity is allowed to adjust inhibitory connections (T1) and, subsequently, plasticity is enabled, when local inhibition reaches a new steady state, we extract activity again to simulate the chronic period of stroke recovery (T2). B) Analysis of modeled data. To accurately represent BOLD signals, model activity from the excitatory populations is passed through a hemodynamic model that mechanistically couples neural activity to the changes in blood oxygenation measured by BOLD fMRI. BOLD signals are then filtered and used to compute measures of connectivity and dynamics. For that, we extract static functional connectivity (FC), functional connectivity dynamics (FCD), the Kuramoto order parameter (KOP) and avalanche size cumulative distribution functions (CDF).

response to an external input $x$ and given by:

$$F(x) = \frac{1}{1 + e^{-\frac{x-\mu}{\sigma}}}, \tag{2}$$

where $\mu$ and $\sigma$ can be understood, respectively, as the excitability threshold and sensitivity of the neural mass response to external input.

**Table 1. Fixed model parameters and ranges of variation of free parameters ($C$, mean delay and $\rho$).**

| Parameter | Value | Units |
|---|---|---|
| $\tau_E$ | 2.5 | ms |
| $\tau_I$ | 5 | ms |
| $c_{EE}$ | 3.5 | - |
| $c_{IE}$ | 3.75 | - |
| $P$ | 0.31 | - |
| $\mu$ | 1 | - |
| $\sigma$ | 0.25 | - |
| $\tau_{homeo}$ | 2500 | ms |
| $C$ | [0.1, 14] | - |
| Mean Delay | [0, 15] | ms |
| $\rho$ | [0.05, 0.3] | - |

The values of the remaining parameters were defined according to [70] and can be consulted in Table 1.

For the given parameters, the local neural mass model behaves as a Hopf-Bifurcation (S1 Fig), switching from a steady state of low activity to oscillations, depending on the level of external input. The frequency of oscillation is controlled by the parameters $\tau_E$ and $\tau_I$. Given that local cortical networks are thought to intrinsically generate gamma oscillations through the interaction between pyramidal cells and fast-spiking inhibitory interneurons [84,85], we chose $\tau_E$ and $\tau_I$ so that isolated neural masses generate oscillations with an intrinsic frequency in the gamma range (~40 Hz) (S1 Fig). The level of input required for the phase transition to occur is, in turn, controlled by $\mu$ and $P$. Therefore, we adjusted these parameters so that an isolated neural mass, with no external input, is poised near the critical bifurcation point and incursions into the oscillatory state occur through the coupling between nodes.

## 2.3. Homeostatic plasticity

We implemented homeostatic plasticity as synaptic scaling of inhibitory synapses [48,64], as it has been shown to take an important part in cortical circuit function and homeostasis [48,66] and has been previously applied in the context of large-scale modeling [68–70,72]. Shortly, local inhibitory weights adapt to maintain excitatory activity ($r^E$) close to a given target firing rate ($\rho$). Therefore, the dynamics of local inhibitory couplings $c_{EI,i}$ are described by the following equation, following [48]:

$$\tau_{homeo} \frac{dc_{EI,i}}{dt} = r_i^I (r_i^E - \rho) \tag{3}$$

where $\tau_{homeo}$ is the time constant of plasticity. Such homeostatic plasticity mechanisms are known to operate in long timescales of hours to days [60] or even weeks in primates [67]. Here, to keep simulations computationally tractable, we chose $\tau_{homeo}$ to be 2.5s. In fact, since the magnitude of $\tau_{homeo}$ solely controls how fast $c_{EI}$ weights evolve towards a steady state, provided that $\tau_{homeo}$ is sufficiently slow for plasticity to be decoupled from the fast dynamics of local oscillations, $c_{EI}$ weights will stabilize to nearly exactly the same values (S2 Fig).

## 2.4. Hemodynamic model

From the raw model activity, we extracted simulated BOLD signals by using a forward hemodynamic model [86], as described in [87]. In short, the hemodynamic model describes the

coupling between the firing rate of excitatory populations ($r^E$) and blood vessel diameter, which in turn affects blood flow, inducing changes in blood volume and deoxyhemoglobin content, thought to underlie the BOLD signals measured through fMRI. A detailed description of the system, explaining the hemodynamic changes in node $i$, is given by:

$$\frac{\delta s_i(t)}{\delta t} = r_i - k_i s_i - \gamma_i(f_i - 1)$$

$$\frac{\delta f_i(t)}{\delta t} = s_i$$

$$\tau_h \frac{\delta v_i(t)}{\delta t} = f_i - v_i^{1/\alpha}$$

$$\tau_h \frac{\delta q_i(t)}{\delta t} = \frac{f_i(1 - (1 - \rho_h)^{1/f_i})}{\rho_h} - \frac{v_i^{1/\alpha} q_i}{v_i}$$

$$y_i = V_0\left(7\rho_i(1 - q_i) + 2(1 - \frac{q_i}{v_i}) + (2\rho_i - 0.2)(1 - v_i)\right) \tag{4}$$

where $y_i$ represents the BOLD signal from node $i$. The parameters were taken from [87]. After passing model activity through the hemodynamic model, the output is downsampled to a sampling period of 0.72s to equate modeled signals to the empirical data obtained from human controls used for model optimization.

## 2.5. Model optimization

Model optimization was performed by considering the global coupling ($C$), mean delay and target firing rate ($\rho$) as free parameters. Similarly to previous studies [70], we represent conduction speeds through the mean of the correspondent conduction delays ($\tau_{ij}$). The range of variation for each of the free parameters is described in Table 1. Within the respective ranges, we selected 25 logarithmically spaced values for $C$, 26 values for $\rho$ in steps of 0.01 and 16 mean delays in steps of 1 ms. During simulations, we record $c_{EI}$ weights every 10s due to their slow evolution and to avoid dealing with large datasets. To ensure that $c_{EI}$ reached a stable or quasi-stable steady state, we ran models for 500 minutes of simulation time or until local inhibitory weights had converged to a steady state, through the test condition described in the supplementary material (S3 Fig, S1 Text). After this stabilization period, homeostatic plasticity was disabled and model activity was recorded for 30 minutes. Similarly to [70], we disable plasticity during the recording of signals to ensure that our final measure of activity is not affected by changes in local synaptic weights, although the slow dynamics of plasticity are unlikely to interfere with the fast dynamics of neural activity.

To evaluate model performance against empirical data, we make use of the following properties of FC, following [83] (Fig 1B):

- **Static FC:** 78×78 matrix of correlations between BOLD time series across all network nodes. Modeled FC matrices were compared with group-averaged empirical FC by computing the correlation coefficient and mean squared error between their upper-triangular elements.

- **FC Dynamics (FCD):** matrix of correlations between the upper-triangular part of FC matrices computed in windows of 80 samples with 80% overlap. Model results are compared to

empirical data by performing a Kolmogorov-Smirnov test between the distributions of values in the respective FCD matrices.

## 2.6. Stroke simulation protocol

To compare cortical activity and networks pre-stroke, post-stroke acute and post-stroke chronic, we implement the following protocol (Fig 1A,ii). First, we initialize the model with optimized hyper-parameters ($C$, $\rho$ and mean delay) and without homeostatic plasticity. We fix the $c_{EI}$ weights to the steady-state values corresponding to that combination of parameters, as obtained from the model optimization procedure, and record 30 minutes of pre-lesion baseline activity (T0). Then, we simulate cortical gray-matter lesions by removing all the connections to and from a single node in the network, similar to previous approaches [68,88]. Without turning homeostatic plasticity on, we extract 30 minutes of simulated activity to represent cortical activity during the acute post-stroke period (T1). Given the slow timescales of homeostatic plasticity in the cortex of primates [67], it is unlikely that the human cortex is able to fully adapt to the post-stroke loss in excitation during the acute period. Therefore, we argue that it is reasonable to simulate it by measuring activity in a lesioned model without homeostatic compensation. We then allow Eq (3) to change $c_{EI}$ weights and simulate a maximum of 500 extra minutes of simulated time or until $c_{EI}$ weights reach a new steady state, using the method described in the supplementary material (S3 Fig). Plasticity is then disabled and 30 minutes of simulated activity are extracted to represent the chronic period of stroke recovery.

In all simulations, Eqs (1) and (2) were solved numerically, using the Euler method with an integration time step of 0.2ms (5kHz). Model simulations and subsequent analysis were implemented in Python using in-house scripts, accessible in https://gitlab.com/francpsantos/stroke-e-i-homeostasis.

## 2.7. Analysis of network dynamics

**2.7.1. Synchrony and metastability.**   To evaluate the effect of stroke on the network dynamics of our model we measured synchrony and metastability (Fig 1B). To do that, we first compute the Kuramoto order parameter (KOP) [89,90], which represents the degree of synchrony among a set of coupled oscillators at a given point in time. The KOP can be calculated as:

$$Z(t) = R(t)e^{i\Phi(t)} = \frac{1}{N}\sum_{n=1}^{N} e^{i\theta_n(t)} \tag{5}$$

where $\theta_n(t)$ represents the instantaneous Hilbert phase of a given node $n$ at time $t$. Synchrony and metastability are defined as the mean and standard deviation of $R(t)$ over time, respectively. Therefore, while synchrony represents the degree of phase coupling between nodes in the network, metastability represents the level of flexible switching between a state of synchrony and asynchrony [90].

**2.7.2. Criticality.**   In critical systems, the size of population events will follow a power-law distribution. In neural systems, such events have been related to neuronal avalanches, where the activation of one of the network elements triggers a response of other elements, until activity dies out. It has been shown that the size and duration of such neuronal avalanches follow a power-law distribution with exponent -1.5 [54,91], at various levels, from local networks to large-scale activity [54,92]. Importantly, it is thought that neural systems may operate at this point of criticality to optimize several network functions, from dynamic ranges to information storage and transmission [57–59,93,94].

To detect neural avalanches in our data, we employ the method from [69]. After time-series from each excitatory node are Z-scored ($E_i(t) = \frac{1}{\sigma(E_i)}(E_i - \hat{E}_i)$), we detect incursions beyond a threshold of ±2.3, thus identifying events that are distinct from noise with a probability of p<0.01. Then, we define events as the time points where the signal first crossed the threshold and avalanches as continuous periods of time where events occurred in the network. Subsequently, to measure criticality, we employ the method developed by [59], comparing the distribution of avalanche sizes in neural data with a truncated power-law with exponent -1.5. Shortly, we computed the measure $k$ using:

$$k = 1 + \frac{1}{m}\sum_{n=1}^{m}(F^{NA}(\beta_n) - F^{PL}(\beta_n)),$$ (6)

where $m = 10$ is the number of logarithmically spaced points $\beta_n$ between the minimum and maximum avalanche sizes, $F^{PL}$ is the cumulative distribution of a -1.5 exponent power-law, truncated so that the maximum avalanche size is the number of nodes in our model ($N = 78$), and $F^{NA}$ is the cumulative distribution of avalanche sizes in the model data. Therefore, a score of $k$ close to 1 means that the system is close to criticality, while scores below and above 1 are characteristic of sub and supercritical systems, respectively.

## 2.8. Analysis of functional connectivity properties

**2.8.1 FC distance.**  To measure the dissimilarity between FC matrices at T0, T1 and T2, we make use of a metric we call FC distance, following [68], defined as the Frobenius norm of the difference between two matrices.

$$distance(FC_1, FC_2) = \sqrt{\sum_i\sum_j(FC_2 - FC_1)_{ij}^2}$$ (7)

**2.8.2 Correlation between FC and SC.**  Given the results of [20], showing a decoupling between functional and structural connectivity in stroke patients, correlated with motor function, we test this biomarker at T0, T1 and T2 by computing the Pearson's correlation coefficient between the upper triangles of FC and SC matrices.

**2.8.3. Modularity.**  Modularity measures the degree to which a network follows a community structure, with dense connections within modules and sparser ones between them. Modularity (Q) was calculated using the formula defined in [22]:

$$Q = \sum_{u \in M}[e_{uu} - (\sum_{v \in M}e_{uv})^2],$$ (8)

where M is a set of non-overlapping modules (groups of nodes) in the network and $e_{uv}$ is the proportion of edges in the network that connect nodes in module $u$ with nodes in module $v$. Similarly to [22], we chose network modules *a priori* to avoid biasing the modularity measure by directly using a clustering algorithm that optimizes community structure in data and also to avoid the problem of varying numbers of modules when using community detection algorithms in data from different time points in the simulation protocol. In our analysis, instead of relying on a pre-defined set of communities, we extract our modules from the empirical FC data, by using a clustering algorithm to detect resting state networks [95]. Shortly, we applied k-means clustering (k = 6) 200 times on the empirical averaged FC matrix and recorded the number of runs each pair of nodes were grouped together in an association matrix. Afterward, we applied k-means clustering (k = 6) to the association matrix to detect modules that could be equated to known resting state networks (S4 Fig). Those networks were then used as

modules for the calculation of modularity. Different clustering algorithms were applied, leading to qualitatively similar results (S4 and S5 Figs). The same was observed for different number of clusters (S5 Fig). Since the formula used for modularity relies on the assumption that graphs are undirected and unweighted, FC matrices were transformed into unweighted graphs by applying a density threshold, through which only a percentage of strongest connections are kept and considered edges of the unweighted FC graph (Fig 1B). Nonetheless, results remain qualitatively very similar when computing modularity in the weighted matrices (S5 Fig). Lesioned regions were removed from the network before computing modularity, similarly to [22].

**2.8.4. Small world coefficient.** The small-world (SW) coefficient measures the degree to which a given graph has small-world properties, i.e. its small-worldness. In SW networks, most nodes are not connected but can be reached from any starting point through a small number of edges. SW coefficients were calculated using the following equation [22,96]:

$$SW = \frac{C/C_{rand}}{L/L_{rand}} \tag{9}$$

where $C$ is the average clustering coefficient of a given graph and $L$ is its characteristic path length. Clustering coefficients measure the degree to which the neighbors of a node are interconnected, and the characteristic path length represents the average of shortest path lengths between all nodes in a graph. Both metrics were computed using the *networkx* module in Python [97]. While $C$ and $L$ represent the values from our simulated data, $C_{rand}$ and $L_{rand}$ represent the same metrics taken from a random unweighted and undirected graph with the same edge density as the FC graphs from simulated data. To account for the intrinsic stochasticity in the process, for each simulated FC matrix, $SW$ was calculated 100 times for different generated random networks and the results were averaged to obtain the final $SW$ value. Similarly to modularity, $SW$ was calculated after applying a density threshold to FC matrices and lesioned nodes were removed before the calculation.

Here, both for modularity and small-world coefficients, instead of performing analysis for edge density thresholds between 4 and 20%, following [22], the range was extended to 40%. This is due to the smaller size of our network (78 vs. 324 brain regions), often leading to unconnected graphs when applying thresholds lower than 20%. While this would not affect the calculation of modularity, the computation of small-world coefficients requires graphs to be connected to calculate average shortest-path lengths. Nonetheless, modularity results are qualitatively similar when performing analysis within the 2–20% range (S5 Fig).

## 3. Results

### 3.1. Model results capture healthy FC data in parameter region corresponding to rich network dynamics

To find the optimal working point of our model that best represents empirical FC and FCD, we ran simulations for all the combinations of global coupling (*C*), mean delay and target firing rate (ρ) described in the methods section. In Fig 2A, we represent the results of model optimization for mean delay = 4 ms, for simplicity of representation. The results of optimization for the remaining combinations of parameters can be consulted in S6 Fig. From these, it can be visualized that 4ms is generally the optimal mean delay, in particular regarding an accurate representation of empirical FCD. Focusing on FC, it can be observed from Fig 2A that the improved fitting is achieved for high couplings and a target firing rate close to 0.2. In addition, the model captures the overall structure of empirical FC (as measured through the correlation coefficient) as well as the magnitude of connectivity (as measured through the MSE) (Fig 2A and 2B). Regarding FCD, there is a wide region in the parameter space where the distribution

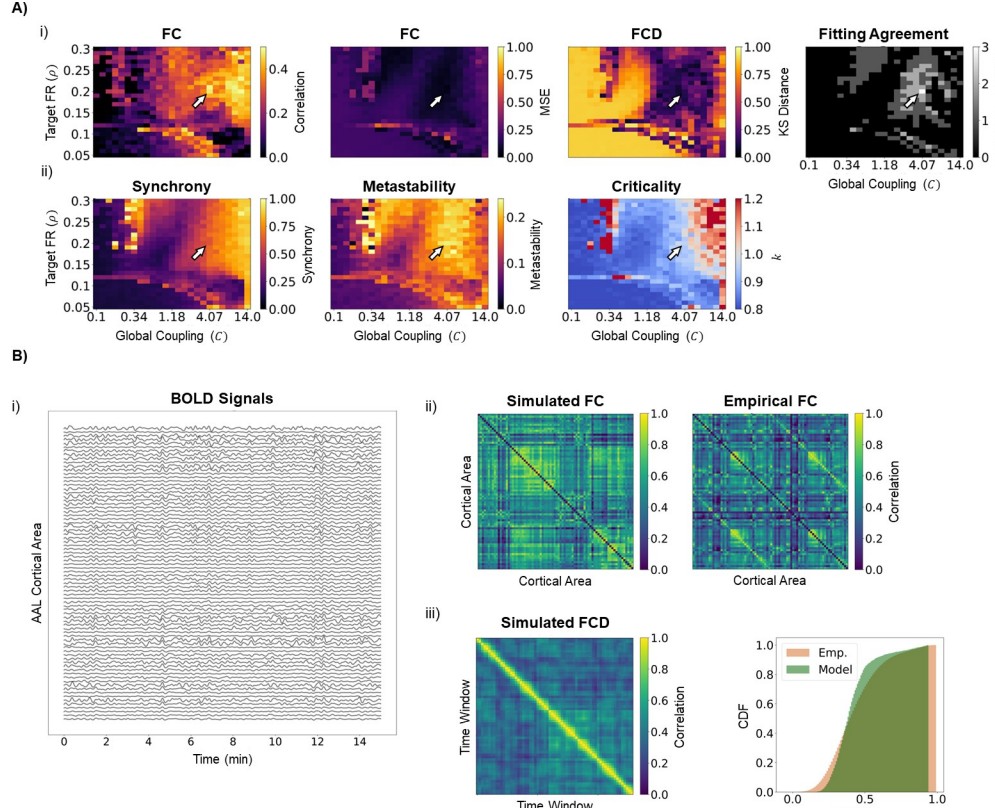

**Fig 2. Model optimization and dynamics.** A) Model fit and dynamics over parameter space. i) Model fit to empirical FC data. The plots represent the results of a grid search over the parameters of global coupling ($C$) and target firing rate (FR) ($\rho$), with the mean delay fixed at 4ms. Model performance was evaluated by the following metrics: (first) Pearson's correlation between the upper triangle of simulated and empirical FC matrices, (second) mean squared error (MSE) between simulated and empirical FC matrices and (third) Kolmogorov-Smirnoff (KS) distance between the distribution of values in simulated and empirical FCD matrices. The rightmost plot shows the result of applying the following thresholds: correlation coefficient $\geq 0.45$, MSE $\leq 0.1$, KS distance $\leq 0.15$. Arrows show the model working point used in the simulations ($C = 4.07$; $\rho = 0.2$; mean delay = 4ms), which satisfies the thresholds for all fitting metrics (correlation coefficient = 0.487, MSE = 0.046, KS distance = 0.138). ii) Model dynamics over parameter space. The plots represent relevant dynamic features of model activity over the explored parameter space: (first) synchrony and (second) metastability representing, respectively, the mean and standard deviation of the KOP over time, and (third) global criticality. Note that the chosen working point is poised in a region of transition between low and high synchrony (synchrony = 0.606), high metastability (metastability = 0.230) and transition between sub and supercriticality ($k = 0.960$). B) Model behavior at the chosen working point. i) Example of 15 minutes of model activity. Note the emergence of transient patterns of co-activation between different areas in the network. ii) Simulated (left) and empirical (right) FC matrices. While generally overestimating connectivity, the model is able to capture empirical FC patterns. iii) Simulated FCD matrix (left) and its cumulative distribution function, compared to the one from empirical data (right).

of modeled FCD matrices matches empirical results. Since the same wide parameter region is not observed for other mean delays (S6 Fig), results suggest that axonal conduction velocity has a significant influence on the accurate representation of FCD in our model. Furthermore, there is a narrow parameter region where we can simultaneously optimize the representation of both FC and its dynamics (Fig 2A,i). In this parameter region, BOLD signals show rich dynamics, characterized by transient co-activation of groups of nodes in the network (Fig 2B), as is characteristic of resting-state cortical signals [98]. Importantly, and following previous studies [99], the optimal region lies in the transition between low and high synchrony, corresponding to a region of optimal metastability (Fig 2A,ii). In addition, this parameter region

further corresponds to global dynamics close to criticality, consistent with previous studies showing that criticality is a property of large-scale cortical networks [92], also observed in models with similar homeostatic mechanisms [69]. Therefore, the model can reproduce, to some level, the structure of FC and its transient dynamics, and is in accordance with the current knowledge of the dynamic features of brain activity. Given these results we choose the following hyperparameter values for the simulations in the subsequent sections, $C = 4.07$, $\rho = 0.2$ and mean delay = 4 ms, as indicated by the white arrows in Fig 2A. Finally, while these results were obtained using an averaged FC matrix and the aggregated FCD distribution across subjects, individual level fitting using subject-specific FC matrices and FCD distributions yields a distribution of optimal points with a dense concentration around our chosen optimal point (S7 Fig).

## 3.2. Excitatory-inhibitory homeostasis contributes to the recovery of static properties of FC

To evaluate the acute effects of lesions in cortical FC and the putative role of E-I homeostasis on its long-term recovery, we simulated cortical lesions by removing all the connections to and from a single node. This was done individually for all the nodes in the network and FC was extracted pre-lesion (T0), immediately after lesion application, an equivalent of the acute period (T1), and after local inhibitory weights reach a new steady state through local E-I homeostasis, which we equate to the chronic period of stroke recovery (T2) (Fig 1A,ii).

When looking at the differences in FC between T1 and T2 (Fig 3A), it can be first observed that, similarly to what occurs in stroke patients, different lesions have highly heterogeneous acute effects. In Fig 3A we represent the strongest 10% changes in FC for lesions in nodes with different strengths (i.e. sum of incoming structural connectivity weights): the right superior frontal gyrus (strength = 6.23), left precentral gyrus (strength = 3.23) and left parahippocampal gyrus (strength = 0.42). Some qualitative conclusions can be drawn from looking at the observation of acute effects of such lesions. First, while there seems to be a general effect of global disconnection (Fig 3A,i and iii), also evident in the median changes over lesions (Fig 3B,i), certain lesions can lead to hypersynchrony (Fig 3A,ii), as previously reported in lesioned brain networks [27,100]. Second, lesions to high degree nodes (Fig 3A,i and ii) have stronger acute effects than lesions to low degree nodes (Fig 3A,iii). Third, different lesions show different levels of recovery in the chronic period (T2), as evidenced by the ipsilesional hypersynchrony observed after lesion in the left precentral gyrus, which was not diminished significantly at T2 in our simulations (Fig 3A,ii). Fourth, regarding the median effects over lesions (Fig 3B,i), we observed a widespread increase in functional connectivity, compared to pre-lesion levels, in a process that could be understood as a global cortical reorganization. More specifically, it is likely that, given the inability to recover connectivity between certain brain areas, new functional connections are formed (or previous ones strengthened) to maintain relevant graph properties of FC. More specifically, the effects of a lesion can be summarized, in a more general way, as follows: a strong acute disconnection, stronger in the ipsilesional side, but extending to the contralesional cortex, as is characteristic of diaschisis [14], and a chronic increase in connectivity, spread across both hemispheres, likely related to the functional reorganization of cortical networks. More specifically, when relating lesion-specific changes in connectivity with healthy FC and SC, we observe that the evolution of FC from T1 to T2 generally counteracts the acute deficit felt at T2 (S8 Fig), indicating a tendency toward adequate recovery. In addition, this follows recent results suggesting that connections that undergo more significant changes during recovery are linked to nodes with stronger acute disruptions in FC [101]. Interestingly, the change in connectivity from T0 to T2 is generally negatively correlated with

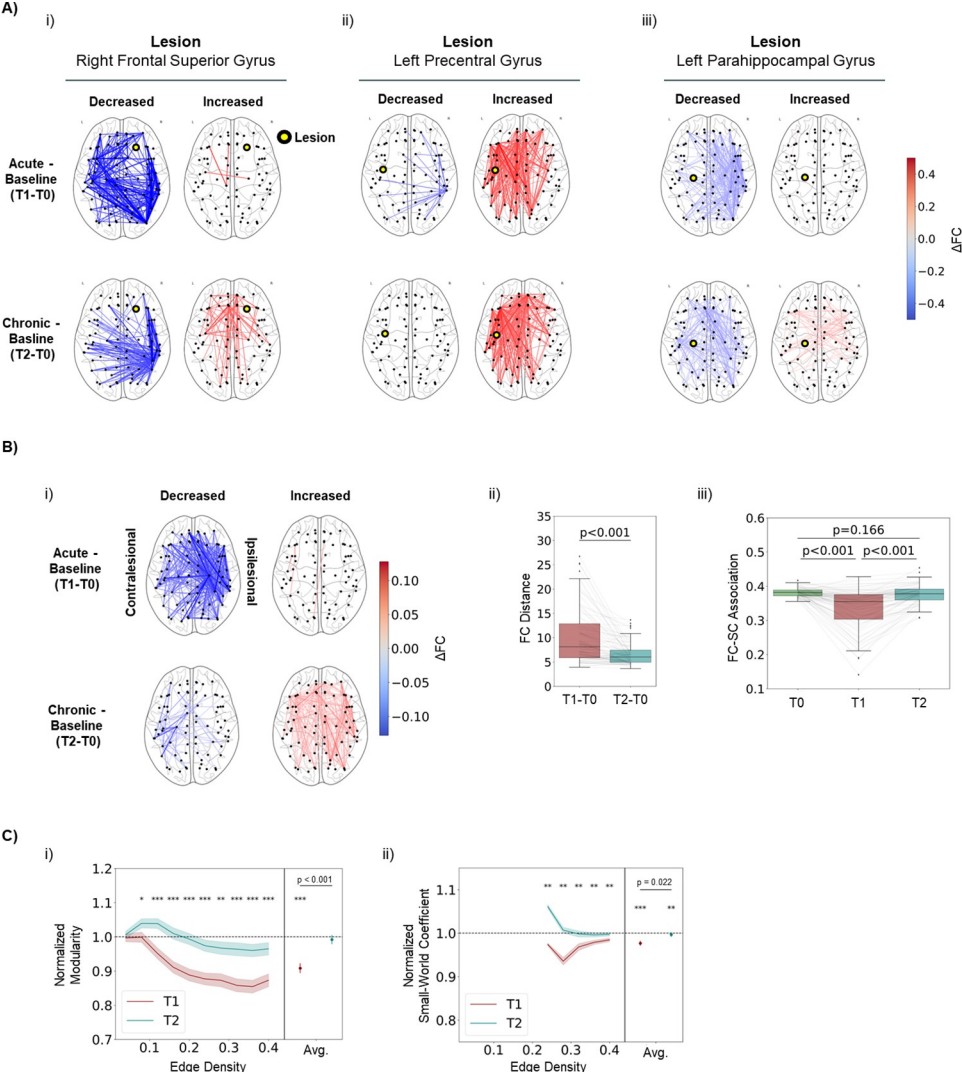

**Fig 3. E-I homeostasis contributes to the recovery of static FC properties.** A) Differences in FC following example lesions in the acute (T1-T0) and chronic (T2-T0) periods. Only the 10% strongest changes are shown. i) Effects of a lesion in the right frontal superior gyrus. ii) Effects of a lesion in the left precentral gyrus iii) Effects of a lesion in the left parahippocampal gyrus. B) Effect of lesion in static properties of FC. i) Median differences in FC over lesions in the acute (T1-T0) and chronic (T2-T0) periods. Data from left-side lesions was mirrored so that the right side was always contralesional. Only the 10% strongest differences are shown in the plot. Note the general disconnection in the acute period, stronger on the ipsilesional side, followed by a widespread increase in connectivity in the chronic period. ii) Distance between FC matrices at T1 and T0, and T2 and T0. FC distance was significantly decreased from the acute (10.202±5.838) to the chronic period (6.664±5.838) (p < 0.001, Mann-Whitney U-test). iii) Pearson's correlation coefficient between the upper triangle of functional and structural connectivity matrices at T0, T1 and T2. We observe a significant decrease from T0 to T1 (p < 0.001, Mann-Whitney U-test) and a subsequent increase towards pre-lesion levels from T1 to T2 (p < 0.001, Mann-Whitney U-test). Results at T0 and T2 were not significantly different (p = 0.166, Mann-Whitney U-test). C) Effect of lesion in graph properties of FC. i) Modularity at T1 and T2, normalized to T0 values, for different edge density thresholds. Lines represent the mean over lesions and shaded areas represent the standard error of the mean. On the right side of each plot, we show results averaged over all edge density thresholds for each lesion. We observed a significant decrease in modularity at T1 (0.908±0.120, p < 0.001, Wilcoxon ranked-sum test), with a significant increase between T1 and T2 (p < 0.001, Mann-Whitney U-test). Normalized modularity at T2 was not significantly different from baseline (0.992±0.110, p = 0.500, Wilcoxon ranked-sum test). ii) Same, for small-world (SW) coefficients. Values show significantly decreased SW coefficients at T1 (0.977±0.043, p < 0.001, Wilcoxon ranked-sum test), with a significant increase between T1 and T2 (p = 0.022, Mann-Whitney U-test). In this case, although values at T2 were close to the baseline, a significant difference could still be observed (0.997 ±0.041, p = 0.007, Wilcoxon ranked-sum test). In both plots, asterisks represent the level of significance of a Mann-Whitney U-test. * p<0.05, ** p<0.01, *** p<0.001.

healthy FC (T0) (S8 Fig), suggesting that the process of reorganization can be understood as formation of new functional connections, as opposed to the further strengthening of previously strong ones.

To measure lesion effects more quantitatively, we measured the distance between FC matrices at T1 and T2 versus T0 across lesions (Fig 3B,ii). It can be observed that there is a strong departure from pre-lesion FC at T1 (FC distance, 10.202±5.838), significantly reduced at T2 (6.664±5.838, p<0.001, Mann Whitney U-test), thus showing a recovery of FC towards pre-lesion patterns. Nonetheless, a difference remains at T2, compared to T0, likely resulting from functional reorganization. Similarly to [68], we found a correlation between graph properties of lesioned nodes and FC distance (S9 Fig), emphasizing the point that lesions in high degree nodes, or structural hubs, cause larger disruptions on FC.

In addition, a decoupling between functional and structural connectivity has been observed in stroke patients and shown to correlate with motor function [20]. Our results replicate this finding in the acute period (Fig 3B,iii) where the average correlation significantly dropped from 0.381±0.013 at T0 to 0.334±0.060 at T1 (p <0.001, Mann-Whitney U-test). Furthermore, similarly to FC distance, we found a correlation between the magnitude of this change and the lesion properties (S9 Fig). Importantly, structural-functional coupling was recovered to pre-lesion levels at T2 (0.376±0.028, T0 vs T2 p<0.001, Mann-Whitney U-test), further indicating the ability of E-I homeostasis to participate in the recovery of FC.

Beyond such metrics of damage to FC, it is relevant to measure changes in graph properties that are relevant in human brain networks, such as modularity [25] and small-worldness [26,96]. More importantly, those were shown to be affected by stroke and, in the case of modularity, to be a robust biomarker of performance in higher-order functions (e.g. memory, attention) [22]. For most of the density thresholds explored, we observed a decrease in modularity at T1, further recovered towards pre-lesion levels at T2 (Fig 3C,i, values normalized to T0). When averaging the values over all the thresholds for each lesion simulation (Fig 3C,i right) we observed a significant decrease in modularity at T1 (0.908±0.120, p < 0.001, Wilcoxon ranked-sum test), further recovered towards baseline at T2 (p < 0.001, Mann-Whitney U-test), with no significant difference from baseline found at this time point (0.992±0.110, p = 0.500, Wilcoxon ranked-sum test). As opposed to FC distance and association with SC, disruptions in modularity did not correlate significantly with the properties of lesioned nodes (S9 Fig). Similarly to modularity, SW coefficients were significantly decreased at T1 (0.977±0.043, p<0.001, Wilcoxon ranked-sum test) and further increased from T1 to T2 (p = 0.022, Mann-Whitney U-test) (Fig 3C,ii),. However, in this case, a significant difference from baseline could still be found at T2 (0.997±0.041, p = 0.007, Wilcoxon ranked-sum test). Note that SW coefficients could only be systematically calculated across lesions for edge density thresholds larger than 20%. Due to the small size of our network (78 nodes), thresholding with smaller edge densities leads to disconnected graphs, on which is not possible to calculate SW coefficients reliably. Nonetheless, besides replicating the acute decreases in modularity and small-worldness found by [22], we further show that E-I homeostasis participates in the recovery of these graph properties, offering a possible explanation for the long-term recovery of such properties reported by the same authors.

To summarize, our results show the strong effect that lesions to the neocortex have on the static properties of FC, and how E-I homeostasis drives their further long-term recovery. While these effects were heterogeneous across lesions, there was a tendency of cortical networks to display a loss in modularity and small-worldness, two relevant properties of cortical function shown to be affected in stroke patients. Such metrics were, however, recovered in the chronic period, likely through functional reorganization, showing the vital role of E-I homeostasis in their recovery.

### 3.3. Excitatory-inhibitory homeostasis is not sufficient for the reinstatement of rich networks dynamics

Beyond post-stroke disruptions in the static properties of functional connectivity, it is relevant to analyze how it affects cortical dynamics. Healthy resting-state cortical activity displays rich spatiotemporal dynamics, with transient activation of distributed networks, jumps from asynchronous to synchronous states [99] and a scale-free distribution of network events of co-activation (i.e. criticality) [69,92]. Therefore, we measure the acute effects of lesions on such dynamical properties and evaluate the possible role of E-I homeostasis in the recovery of dynamical features that go beyond static FC networks.

If Fig 4A,i, we plot the distribution of FCD values at T0, T1, and T2 for the same example lesions described in the previous section. Although some level of heterogeneity can be found across lesions, the general effect, further visible in the distribution of FCD values across lesions (Fig 4A,ii), is a shift towards higher values at T1, which could not be recovered at T2. Such a shift is difficult to interpret, due to the lack of similar analyses in literature. However, given the definition of FCD values as the correlation between FC taken from different time windows in the signal, a functional interpretation can be given. Such a shift could mean that transient FC motifs were more similar across time, indicating a more rigid spatiotemporal pattern of activation, likely due to a loss in the richness of dynamics previously described. However, functional interpretations should be taken with careful consideration, given the lack of empirical studies debating the effects of stroke in FCD and its clinical correlates. Nonetheless, looking at other dynamical properties might shed light on the issue. Regarding synchrony (Fig 4B,i), we observed highly heterogeneous effects, similar to the previous modeling study of [27], where networks can change to either increased or decreased synchrony, in line with the results of the previous section (Fig 3A,ii), showing hyperconnectivity in the acute phase post stroke for selected lesions. More importantly, we observed a significant decrease in metastability (Fig 4B, ii) at T1 (-4.932±7.211%, p < 0.001, Wilcoxon Ranked-Sum test) and, while there was a significant shift towards baseline between T1 and T2 (p = 0.008, Mann-Whitney U-test), metastability at T2 was still significantly lower than in the pre-lesion period (-2.144±6.239%, p = 0.004, Wilcoxon Ranked-Sum test). Since high metastability has been associated with the ability of the brain to switch between FC states [99], this might relate to the hypothesized rigidity of FCD patterns (Fig 4A,ii). Therefore, we suggest a decreased flexibility of resting-state dynamics in stroke patients. In addition, while dynamics at T0 were found to be close to criticality (Fig 4B,iii) ($k = 0.972±0.022$), we observed a significant shift towards sub-criticality at T1 ($k = 0.948±0.034$, p<0.001, Mann-Whitney U-test). Importantly, dynamics were still significantly sub-critical compared to T0 ($k = 0.950±0.025$, p<0.001, Mann-Whitney U-test), with no significant recovery occurring between T1 and T2 (p = 0.935, Mann-Whitney U-test). Therefore, the overarching conclusion from the analysis of dynamics in our simulations is that stroke lesions have a strong effect on network dynamics and, more specifically, in metrics that can be understood as quantifying rich network dynamics, such as metastability [99] and criticality [92]. More importantly, as opposed to the static properties of FC, the affected dynamics could not be recovered through the E-I homeostasis mechanism implemented in our model, showing a higher fragility of cortical dynamics to stroke, when compared to connectivity.

### 3.4. Long-term changes in local excitability replicate empirical findings from stroke models and patients

While previous studies have attempted to model similar E-I homeostasis mechanisms to assess their relevance in post-stroke recovery [68], we extend our analysis by systematically assessing the changes in local excitability required to adapt to the post-lesion loss in excitation and how

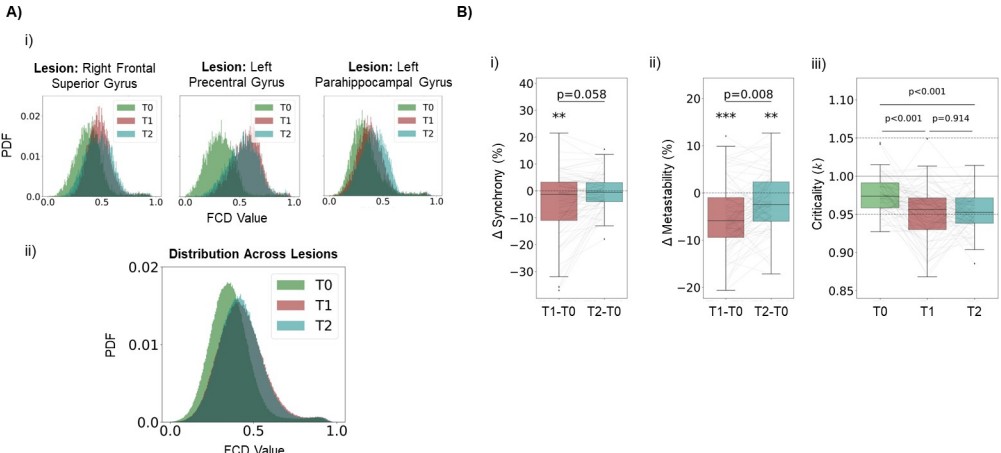

**Fig 4. E-I homeostasis is not sufficient to recover features of rich dynamics.** A) Effects of lesion in FC dynamics. i) Distribution of values in FCD matrices for T0, T1 and T2 for lesions in Right Frontal Superior Gyrus (left), Left Precentral Gyrus (middle) and Left Parahippocampal Gyrus (right). ii) Distribution across lesions of values in FCD matrices for T0, T1 and T2. Note the shift towards higher values at T1 and the similar distribution at T2, denoting an inability of E-I homeostasis to return FC dynamics to pre-lesion levels. B) Effects of lesion in network dynamics. i) Changes in synchrony, in percentage, at T1 and T2, compared to baseline (T0). While synchrony showed a significant decrease at T1, (-4.743±12.288%, p = 0.007, Wilcoxon Ranked-Sum test), there was no significant difference between values at T1 and T2 (p = 0.058, Mann-Whitney U-test). In addition, the difference in synchrony at T2 was not significantly different from 0 (-0.187±6.489%, p = 0.058, Wilcoxon Ranked-Sum test). ii) Same, for metastability. We observed a significant decrease at T1 (-4.932±7.211%, p < 0.001, Wilcoxon Ranked-Sum test), further recovered towards pre-lesion levels at T2 (p = 0.008, Mann-Whitney U-test). However, metastability at T2 was still significantly different from baseline at T2 (-2.144±6.239%, p = 0.004, Wilcoxon Ranked-Sum test). iii) Criticality at T0, T1 and T2. We observed a shift towards subcriticality at T2 (p < 0.001, Mann-Whitney U-test), with no recovery from T1 to T2 (p = 0.914, Mann-Whitney U-test).

they distribute across the brain. We do this by looking at the change, from T0 to T2, in the strength of local inhibitory coupling $c_{EI}$. More specifically, we consider decreases/increases in $c_{EI}$ to represent increases/decreases in excitability, respectively. Importantly, long-term increases in excitability have been found in the cortex of mice models [36,42,43] and stroke patients [38,39,41], mostly related to decreased levels of inhibition. Therefore, it is relevant to evaluate if such effects can, at least to some extent, be a result of physiological processes of E-I homeostasis, tied to the recovery of not only local E-I balance, but also FC properties, as demonstrated by our previous results.

That said, in Fig 5A, we plot the long-term changes in excitability observed across the cortex for the same example lesions referenced before. From these plots, it can be deduced that lesions in more connected nodes required larger changes in excitability. Moreover, the strongest increases in excitability are felt closest to the lesioned areas (Fig 5A,i and ii), as evidenced by previous research in rodent models of stroke [36]. More specifically, for stronger lesions, $\Delta c_{EI}$(%) could be reasonably explained as an exponential function of Euclidean distance to the lesion ($R^2$ = 0.65 and 0.50 for lesions in the right superior frontal gyrus and left precentral gyrus, respectively). This relationship was lost for weaker lesions ($R^2$ = 0.02 for lesion in the right parahippocampal gyrus), likely due to the less widespread and overall weaker effects (Fig 3). We explain these variations as an exponential function of distance given the exponential dependence found between structural connectivity and distance in the cortex [102] and the fact that areas more strongly connected to the lesion would experience the strongest loss in excitation. Therefore, an exponential relationship between $\Delta c_{EI}$(%) and distance to the lesion is almost trivial, as observed for the most severe lesions in our simulations. More specifically, our results suggest that the distance-dependence of changes in excitability is likely a

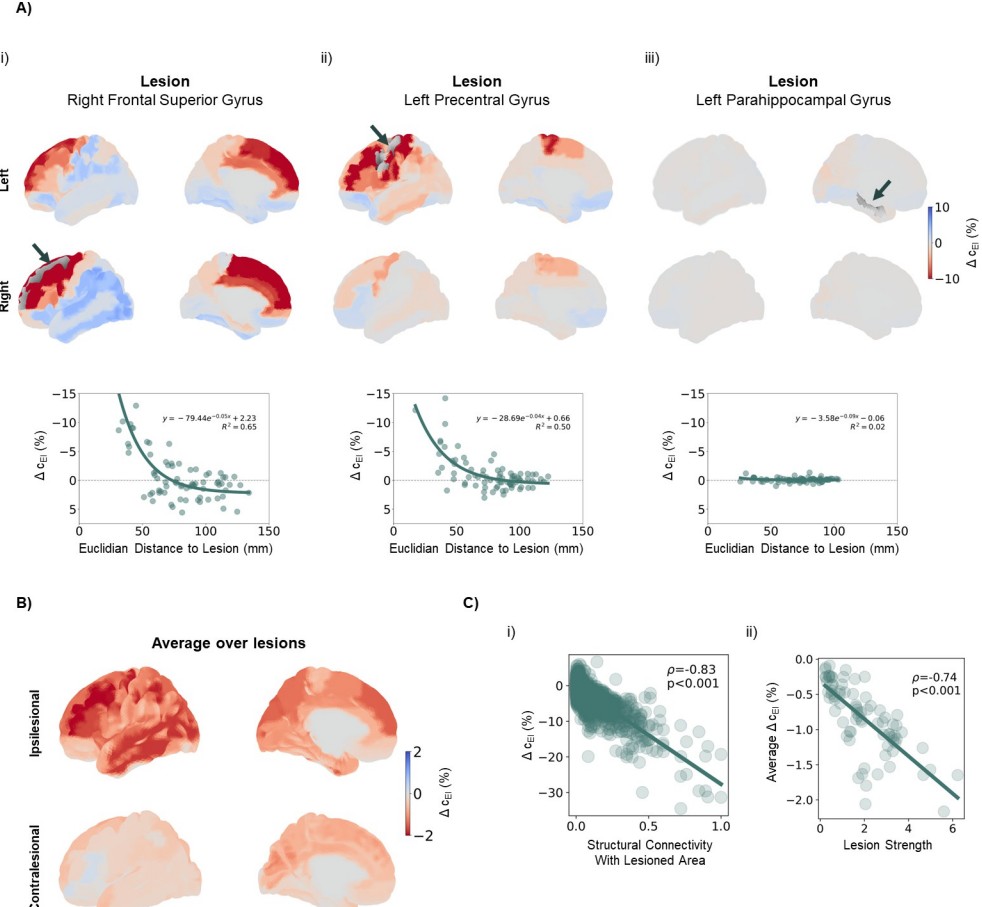

**Fig 5. Long-term adaptations required to recover E-I balance replicate observed post-stroke changes in excitability.** A) Examples of long-term changes in excitability, quantified through the difference in local $c_{EI}$ weights (in percentage) between T0 and T2, in response to different lesions. (Top) Changes in local excitability, projected onto an anatomical map of the human cortex. Red colors represent increases in excitability (decreased inhibition) and blue colors show deceased excitability (increased inhibition). Arrows and gray shading indicate the location of lesioned areas. (Bottom) Changes in excitability against euclidian distance to the lesioned area with results of an exponential fit to the data and respective $R^2$ values. i) Response to a lesion in the right frontal superior gyrus. Note the strong changes across the cortex, with the highest increases concentrated in the vicinity of the lesion, decreasing exponentially with distance ($R^2 = 0.65$) ii) Response to a lesion in the left precentral gyrus. Again, the highest increases in excitability occur close to the lesioned area, with a distance-dependent exponential decay ($R^2 = 0.50$). iii) Response to a lesion in the right parahippocampal gyrus. Note the weaker changes and the poor exponential fit ($R^2 = 0.02$). B) Long-term changes in excitability averaged over lesions. Data from left-side lesions was mirrored so that the right side was always ipsilesional. Note the general increases in excitability across the cortex, strongest on the ipsilesional side. C) Relationship between changes and lesion properties. i) Local changes in excitability against structural connectivity with the lesioned area ($W_{ij}$ where $i$ is the region where $\Delta c_{EI}$ is measured and $j$ is the lesioned area. $\Delta c_{EI}$ correlated strongly with $W_{ij}$ (Pearson's correlation coefficient -0.83, p<0.001 F-test). ii) Average $\Delta c_{EI}$ across cortical areas, plotted against lesion strength (node strength of lesioned areas, $\sum_i w_{ij}$). Average changes were strongly correlated with lesion severity (Pearson's correlation coefficient, -0.74, p<0.001 F-test).

consequence of the underlying organization of structural connectivity, which decays with distance in an exponential manner [102]. On another note, while the consensus in the literature favors a long-term increase in excitability during stroke recovery, we observe, in particular for stronger lesions, actual decreases in excitability in distant cortical regions. This response is likely a second-order effect, resulting from the strong increases in excitability in the areas

closest to the lesion, which in turn might require an opposite reaction in other regions that might be connected to them, but not to the lesioned area itself.

In Fig 5B, we plot $\Delta c_{EI}$(%) averaged across lesions. Here, data from lesions left-side lesions was mirrored before averaging so that the right side always corresponded to the ipsilesional hemisphere. Looking at the average changes across lesions (Fig 5B) shows a picture of widespread increases in local excitability, following literature. Importantly, such increases were significantly stronger (p<0.001, Mann-Whitney U-test) in the ipsilesional cortex (-1.257 ±3.345%), when compared to its counterpart (-0.417±1.212%), as expected due to the distance dependence of changes in excitability. The strongest differences were found in the ipsilesional middle frontal gyrus (-2.205±5.195%), precentral gyrus (-2.144±4.420%), inferior parietal gyrus (-2.100±4.289%), middle occipital gyrus (-1.982±3.179%) and inferior (-1.963±5.632%) and middle (-1.949±4.015%) temporal gyri. However, while we might observe these general effects, the changes in excitability are still highly dependent on the specific lesioned area. In Fig 5C,i, it can be seen that areas with stronger structural connectivity with the lesioned cortex have to undergo higher increases in excitability (Pearson's correlation coefficient = -0.83, p<0.001 F-test), with local changes in $\Delta c_{EI}$(%) being as high as 30%. Moreover, the average increase in excitability across cortical regions shows a relationship with lesion severity (Pearson's correlation coefficient = -0.74, p<0.001 F-test, Fig 5C,ii). Therefore, lesions in well-connected areas require stronger compensation, particularly in nodes that are more strongly connected to the lesion.

In conclusion, by accounting for the participation of slow mechanisms of E-I homeostasis in stroke recovery, we replicate empirical findings in stroke patients and models, such as an overall increase in excitability driven by a decrease in inhibitory transmission [36,38,39] and decaying with distance to the lesion [36]. Moreover, such changes can be predicted for individual cortical areas, given their structural connectivity to the lesioned cortex. It is important, then, to stress that this leads to high heterogeneity in homeostatic changes, showing the importance of developing personalized models where patient-specific information about structural connectivity and damaged areas can be integrated to predict the long-term changes in excitability required for recovery of E-I balance.

## 3.5. Long-term changes in excitability relate to biomarkers of common side-effects of stroke

Stroke patients tend to develop some side effects during the months post-stroke, such as seizures [3–5], depression [8–10] and chronic pain [6,7], among others. Importantly some of these pathologies have been previously associated with altered patterns of excitability in the cortex (e.g epilepsy [103,104], depression [10] and neuropathic pain [105]). Given the widespread changes in excitability presented in the previous section, it is then relevant to investigate a possible relationship between such homeostatic processes, necessary to maintain local E-I balance, and the emergence of long-term side-effects of stroke.

One such side-effect is the occurrence of post-stroke seizures, which affects up to 22% of stroke patients [3]. When such seizures become recurrent, they are classified as post-stroke epilepsy, occurring in about 7% of stroke patients [106]. In addition, the occurrence of seizures or epilepsy has been previously related to hyperexcitability of areas located in the epileptic focus [103,104] and, while the cause of post-stroke seizures is not yet well known, it has been hypothesized that it relates to the increased excitability in a similar manner [3,106,107]. While epileptic foci can be distributed across the brain, the most common location observed in humans is the temporal lobe and, more specifically, the medial temporal gyrus [104,108]. Accordingly,some of the largest average increases in excitability are found in the ipsilesional

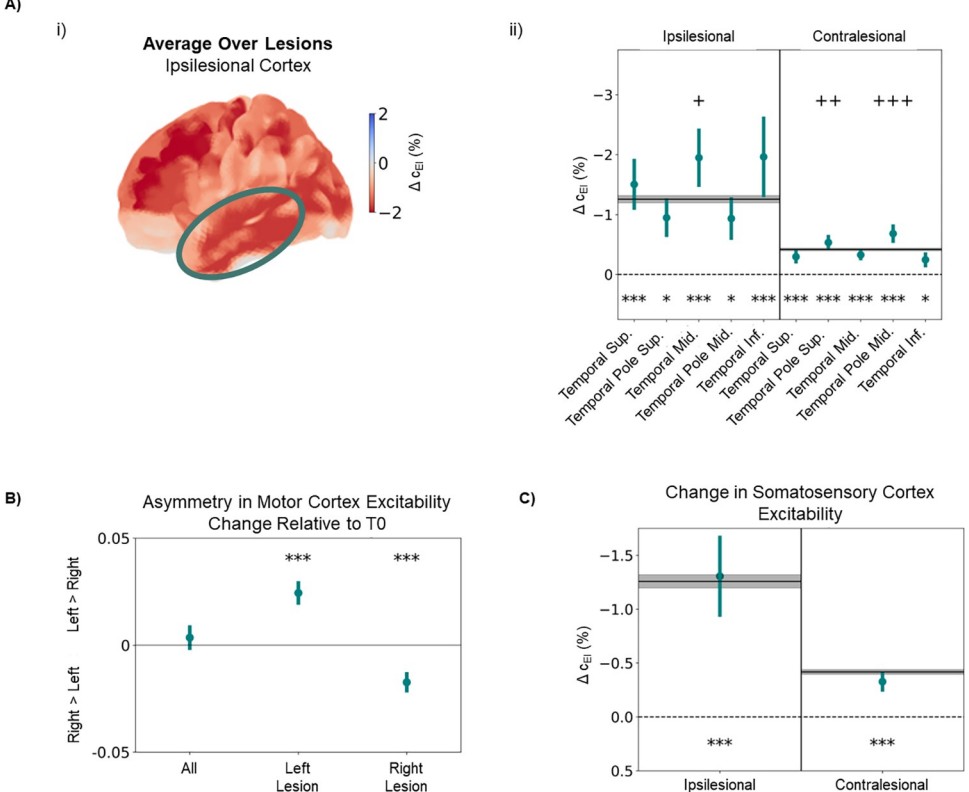

**Fig 6. Long-term changes in excitability relate to known side effects of stroke.** A) Changes in excitability in the temporal cortex, averaged across lesions, quantified through the difference in local $c_{EI}$ weights (in percentage) between T0 and T2 i) Changes in excitability in the ipsilesional cortex. The circled region corresponds to the temporal cortex, where strong increases in excitability can be observed. ii) Changes in excitability for different cortical regions in the temporal lobe, for both ipsi- and contralesional cortex. Black lines and gray shaded areas represent, respectively, the mean and standard deviation of $\Delta c_{EI}$ across all cortical areas. While all regions display a significant increase in excitability between T0 and T2, the ipsilesional middle temporal gyrus, a common location for epileptic foci, showed a significant increase even when compared with the rest of the ipsilesional cortex (p = 0.036, Mann-Whitney U-test). On the contralateral side, while changes are generally weaker, there was a significant difference from the remaining cortical areas in both the superior temporal pole (p = 0.004, Mann-Whitney U-test) and the middle temporal pole (p < 0.001, Mann-Whitney U-test). B) Change in asymmetry between excitability in left and right motor cortices, calculated as the difference, in percentage, of $^{c_{EI,left}}/_{c_{EI,right}}$ from T0 to T2. Note that, for right-side lesions, a change occurs towards higher excitability on the right side (p < 0.001, Wilcoxon ranked-sum test), while the opposite effect is observed for lesions on the left side of the cortex (p < 0.001, Wilcoxon ranked-sum test). C) Changes in excitability in the somatosensory cortex (postcentral gyrus). While changes were not significantly different from the remaining cortical areas, in both ipsilesional (p = 0.702, Mann-Whitney U-test) and contralesional (p = 0.195, Mann-Whitney U-test) cortices, both somatosensory areas underwent significant increases in excitability (ipsilesional: -1.306±0.702, p<0.001; contralesional: -0.327±0.195, p<0.001; Wilcoxon ranked-sum test). In all plots, points represent the average over lesions and bars represent the standard error of the mean. Asterisks represent a significant difference from 0, using the Wilcoxon ranked-sum test. * p<0.05, ** p<0.01, *** p<0.001. Crosses represent a significant difference from the distribution of $\Delta c_{EI}$ across either ipsi- or contralesional cortices, using the Mann-Whitney U-test. + p<0.05, ++ p<0.01, +++ p<0.001.

temporal lobe (Fig 6A, circled area). More specifically, all gyri of the temporal lobe experience significant increases in excitability (asterisks represent the level of significance in a Wilcoxon ranked-sum test), in both ipsi and contralesional cortices. More importantly, the ipsilesional middle temporal gyrus undergoes a particularly strong increase in excitability (-1.949 ±4.015%), significantly larger than the remaining areas in the ipsilesional cortex (p = 0.036, Mann-Whitney U-test). Therefore, the emergence of post-stroke seizures may be potentiated by the action of E-I homeostatic mechanisms, although the magnitude of causality is difficult

to assess. Interestingly, most post-stroke seizures result from cortical lesions [107], precisely the type of lesions applied in our computational model of stroke. Furthermore, it is important to stress again the high heterogeneity of effects over lesioned areas observed in our results. Besides the higher prevalence of post-stroke seizure in patients with cortical lesions, the literature is not clear regarding the location of lesions most likely to lead to this side-effect. Here, we predict that lesions to the temporal cortex would have the highest likelihood of leading to post-stroke seizures, due to the strong connectivity and spatial proximity between temporal areas, which, as shown in the previous section, would lead to higher increases in excitability. Nonetheless, lesions in the angular and middle occipital gyri could also cause strong increases in excitability of the middle temporal cortex (S10 Fig).

Another common side effect of stroke is depression, with an estimated prevalence of 17–52% in stroke patients [8,9]. While some studies argue that the main factors of risk pertain to the social situation of stroke patients, gender and a history of previous depression [109] others suggest a dependence on lesion location, showing a higher prevalence of post-stroke depression (PSD) in patients with right side lesions and lesions in more frontal areas [110]. Furthermore, depression, in particular major depressive disorder, has been associated with asymmetry in cortical excitability [111], particularly between motor cortices and towards higher excitability of the right side [112,113]. Therefore, we hypothesize that, after lesions on the right side, there is an increase in the asymmetry of excitability towards the right motor cortex, when compared to pre-lesion levels. To measure this change quantitatively we compute the following metric, quantifying changes in asymmetry of motor cortex excitability:

$$\frac{c_{EI,right}(T2)/c_{EI,left}(T2)}{c_{EI,right}(T0)/c_{EI,left}(T0)} - 1 \qquad (10)$$

Shortly, if the index is negative, the ratio between the right and left motor cortex $c_{EI}$ weights decreased from T0 to T2, meaning excitability increased more on the right side than on its left counterpart. If Fig 6B, we plot this value over all lesions, and split it between lesions on the left and right sides. While the average over all lesions shows no significant change in motor cortex excitability asymmetry (p = 0.465, Wilcoxon ranked-sum test), for right side lesions we observed a significant shift towards higher excitability of the right motor cortex (p<0.001, Wilcoxon ranked-sum test). This result is, therefore, simultaneously consistent with the observation of this biomarker in depressive subjects [112,113] and with the higher prevalence of PSD in patients with right-side lesions [110]. For left-side lesions, the opposite variation was found (p<0.001, Wilcoxon ranked-sum test). In fact, under the framework of E-I homeostasis, such results are trivial, considering that the ipsilesional cortex tends to experience a higher increase in excitability than its counterpart. Therefore, right-side lesions would lead to a generalized shift in the symmetry of excitability towards the right side, as predicted by our model (Fig 5B). In fact, such asymmetries have been found in human subjects beyond the motor cortex, with studies reporting similar changes in the frontal cortex [111]. Furthermore, results are still heterogeneous across lesions, with the highest changes in asymmetry of motor cortex excitability towards the right side found for lesions in the right superior and medial frontal gyri, right postcentral gyrus, and right paracentral lobule (S11 Fig). The stronger changes observed for lesions in the superior and middle frontal gyrus, in accordance with the higher prevalence of depression in patients with more frontal lesions [110], lend further strength to our hypothesis.

Another common post-stroke side effect is neuropathic pain, occurring in 11–55% of stroke patients, although not always associated with the stroke itself [7,114]. Neuropathic pain is generally hypothesized to relate to an increase in neuronal excitability of somatosensory areas

[115], as is also the case when it occurs post-stroke [7]. This increased somatosensory excitability would then lead to a lower threshold for pain. Such changes are thought to be caused by maladaptive plasticity of the somatosensory cortex [7,115]. In Fig 6C, we plot the change in excitability of the ipsi and contralesional somatosensory cortices (i.e. postcentral gyrus). While none of these areas was related to a significant increase in excitability compared to the rest of the cortex on the same side (ipsilesional: p = 0.702; contralesional: p = 0.195; Mann-Whitney U-test), both the ipsi and contralateral cortices showed a significant increase in excitability from T0 to T2, stronger in the ipsilesional side (ipsilesional: -1.306±3.163%, p<0.001; contralesional: -0.327±0.672%, p<0.001; Wilcoxon Ranked-sum test). Changes were stronger for lesions in the precentral gyrus, superior parietal gyrus, supramarginal gyrus and paracentral lobule (S12 Fig).

Therefore, the changes in excitability operated by E-I homeostasis to adapt to the loss of long-range excitatory input might be involved in the appearance of reported side effects of stroke such as epilepsy, depression and neuropathic pain. However, it is important to stress that it is difficult to estimate the magnitude of causal influence between E-I homeostasis and the incidence of the mentioned side effects, since previous research has highlighted other important risk factors, such as a lack of social support in the case of depression [109]. Nonetheless, E-I homeostasis may inadvertently contribute either to a higher propensity of stroke patients to develop the aforementioned symptoms or to exacerbate their intensity.

## 4. Discussion

We show that our model, by optimizing local and global parameters (i.e. target firing rate and global coupling), can simultaneously represent empirical FC and relevant dynamical features of cortical activity. By simulating cortical stroke lesions, we further show that E-I homeostasis, a mechanism that is well documented in the cortex [60], likely takes part in the recovery of relevant static properties of FC, from FC-SC correlation [20] to complex graph properties such as modularity and small-worldness [22]. Importantly, our framework of E-I homeostasis offers a possible explanation for the emergence of such macroscale properties of cortical activity. We believe that our multiscale approach, bridging the meso (node-level E-I balance) and macro (network level properties) scales of cortical dynamics, offers a relevant perspective on the governing principles of emergence of cortical network features which are still not well understood [116].

Conversely, E-I homeostasis was not sufficient for the recovery of pre-lesion dynamics, such as criticality and metastability, suggesting that, while the global properties of FC can be recovered through local homeostasis of E-I balance, the recovery of dynamics required further adaptive responses from the human cortex. Importantly, we analyze in detail the changes in excitability operated by E-I homeostasis, replicating the known dependence between changes in excitability and distance to the lesion [36]. Here, we bring this further by showing that this dependence is exponential, likely due to the exponential decay of structural connectivity with distance [102]. While the general effect of a widespread increase in excitability is in concurrence with literature [36–39], we stress the high heterogeneity across lesions, with local decreases in excitability observed in particular cases. Importantly, we tie some of the observed changes with biomarkers of known lasting side-effects of stroke, such as seizures [3,5], depression [10,109] and neuropathic pain [7] related to altered patterns of excitability. Therefore, we suggest E-I homeostasis is responsible for either increasing the tendency of stroke patients to develop such side effects, or at least enhancing their effects, while they might emerge from other causes, such as decreases in social activity or intellectual impairment [109].

### 4.1. E-I Homeostasis in stroke recovery

The possibility E-I homeostasis participating in stroke recovery has been suggested before [13,44,45], given the logical association between the acute loss in excitability and the long-term changes in excitability, understood as the subsequent adaptive response from cortical networks to restore E-I balance [60]. In this study, we show that E-I homeostasis can have an important participation in stroke recovery, tying the recovery of global FC properties to local E-I balance. However, one must not neglect the influence of other possible strategies of adaptation, such as structural plasticity [28], vicariation [117] and functional reorganization potentiated by rehabilitation strategies [118,119]. Indeed, it is likely that these processes of recovery interact, since neurostimulation techniques such as theta-burst stimulation, shown to be beneficial for stroke rehabilitation, can simultaneously alter local excitability and long-range functional connectivity [120,121]. It is relevant to stress that the recovery of important properties such as modularity and small-worldness, in our results, is not tied to a full recovery of FC in a connection-by-connection manner. While there is recovery between the acute and chronic periods, FC matrices are still significantly different from baseline in the latter, while the aforementioned properties are mostly reinstated. Therefore, we suggest that, remarkably, the recovery of the graph structure of FC is indirectly orchestrated by local processes of E-I homeostasis and is achieved through a global reorganization of functional connections. This offers an explanation as to why the cortex can coordinate the recovery of such global properties of FC, while individual cortical areas are virtually agnostic to the connectivity (or lack of it) between the remaining cortex. In addition, our results show that such reorganization is generally based in the formation of new functional pathways (i.e. preferential strengthening of the previously weakest connections) and that the greatest changes are also observed in the connections that experience strongest acute disruptions, in line with recent results [101].

### 4.2. Global dynamics of the post stroke brain

Despite the recovery of static properties of FC, our results show a different picture for relevant dynamical features which can be understood as metrics of 'richness' of dynamics. Both metastability, quantifying the ability of a network to flexibly switch between synchronous and asynchronous states [99] or criticality [55], underlying balanced propagation of activity, are significantly affected by lesions and were not recovered solely through E-I homeostasis. A possible explanation would be the fragility of cortical dynamics to disruptions in the structural scaffold of the human cortex, which cannot be compensated solely by local synaptic scaling. Indeed, recent results [28], suggest that, similarly to our results, stroke lesions bring cortical dynamics to subcriticality. More importantly, dynamics could be brought back to criticality in the long-term, but through structural plasticity of white-matter tracts, suggesting that other forms of plasticity beyond synaptic scaling are relevant for the recovery of global dynamics. As for metastability, empirical investigation of its evolution in the brain of stroke patients is lacking. The same is the case for FCD, which measures the transient dynamics of FC. In our results, FCD distributions experience a shift towards higher values, unable to be recovered, similarly to the aforementioned dynamical features. A possible interpretation is a more rigid spatiotemporal pattern of FC, where the cortex has a higher difficulty in switching between different FC patterns associated with the known resting state networks [95]. This might be tied to the decrease in metastability, since rich spatiotemporal FC variation has been hypothesized to be an emergent property of metastable brain dynamics [99]. Relating such results to current knowledge in post-stroke FC dynamics is difficult for a number of reasons. In [31], for example, there is a focus on motor symptoms and the dynamics of somatomotor, subcortical and cerebellar network, which could mask out the dynamics of other cortical networks more

related to cognitive deficits. In [32], where data from the whole cortex was analyzed, the number of state switches did not differ significantly between stroke patients with different levels of impairment, however a comparison to healthy levels was not provided. Nonetheless, the authors indicate an important role of default-mode network dynamics in stroke recovery, which suggests further investigation. In addition, given the relationship between FCD and metastability and higher order cognition [29,30,99,122], we expect that FCD alterations in stroke patients to be more reflective of cognitive deficit. Nonetheless, further results [33], although not directly exploring the ability of the brain to flexibly switch between transient states, indicate that the distribution of state fraction times in severely impaired patients is skewed towards two specific states associated with stroke-related FC deficits. Therefore, the preference of the severely impaired stroke brain for such states might indicate an impaired ability to transition to other configurations associated with healthy dynamics. To explore this issue, we suggest future studies should focus on using methods such as Hidden Markov Modelling [123,124] or leading eigenvector dynamics analysis [122], evaluating the ability of the stroke brain to flexibly transition between states and how it evolves during the process of recovery.

## 4.3. E-I Homeostasis in stroke patients: Recovery time and possible impairments

A relevant reflection to extract from our simulations is the fact that more severe lesions (i.e. lesions in nodes with high connectivity) require a longer time to converge and stabilize (S13 Fig). This result is indeed trivial under the framework of E-I homeostasis, since the evolution of $c_{EI_i}$ after a perturbation in the average firing rate of excitatory populations approximates the behavior of exponential decay (S2, S3 and S13 Figs). Therefore, lesions in nodes that are well connected in the structural network entail a stronger perturbation in the levels of incoming excitation to the rest of the cortex, requiring longer adaptation times. In fact, literature indicates that there is a high level of heterogeneity in the degree of recovery in the first three months post-stroke [125]. While this difference can be attributed to a number of factors, infarct size and location are of particular importance [126]. This is in line with our results, since the graph properties of nodes (S14 Fig) are related to recovery time, which could explain, to some degree, the heterogeneity observed in stroke patients measured at the same instance of the recovery process.

A further consideration from our study is that, in the modeling approach, we assume E-I homeostasis through inhibitory synaptic scaling to be fully functional. While this process has been found to respond robustly to perturbations such as sensory deprivation in rodents [64,66,127], further studies also advance the possibility of impairments in homeostatic plasticity occurring in pathological states [128,129]. Therefore, there is a possibility that E-I homeostasis experiences some level of impairment during stroke recovery. More so, research in homeostatic plasticity suggests that synaptic scaling may not be sufficient to adapt to certain perturbations and that other processes such as regulation of intrinsic excitability might come into play for stronger disruptions [60]. That said, the ability of cortical circuits to homeostatically regulate their own E-I balance may be affected post-stroke, possibly in a patient-specific manner. In fact, literature shows variability in either the strength of inhibition [37,38] or the magnitude of its longitudinal variation in stroke patients [39]. While this variability could be attributed to several heterogeneities between patients (e.g. lesion location, rehabilitation procedures), the strong correlation with behavioral improvement found in [39] suggests that the magnitude of homeostatic adaptation is vital for recovery. We hypothesize that patients with putative impairments in E-I homeostasis would have more difficulty in regaining function.

Importantly, this possibility raises the question of how to modulate cortical circuits to correct such deficits in E-I homeostasis, as has been suggested for the treatment of mood disorders [128]. A possibility is the use of neurostimulation methods, such as theta-burst stimulation, which have been shown to modulate the excitability of cortical areas [121] and that could be applied to specific regions of the cortex undergoing particularly strong increases in E-I homeostasis. Coincidentally, such methods modulate functional connectivity, with effects spreading beyond the stimulated area [120] and, while the precise ties between the modulation of excitability and connectivity are not yet known, such procedures may also stimulate the large-scale reorganization needed to recover the graph-properties of FC. Moreso, since our results show that the magnitude of the adaptive response of E-I homeostasis is related to the graph properties of the connectome and the lesioned areas (Fig 5C), we believe that neuromodulation methods should account for the graph structure of FC, together with the particular disruptions found in a patient-by-patient basis. Indeed, recent results in the context of epilepsy show that the extent of lesions required to stabilize epileptic activity is related to the graph properties of the lesioned areas [130], in line with our results and emphasizing the relevance of accounting for these factors in therapeutical approaches.

An important challenge, then, would be how to detect localized disruptions in E-I balance, i.e. particular regions of the cortex where E-I homeostasis was not able to fully adapt. For example, it is possible that impairments in E-I homeostasis would more likely arise in the vicinity of the lesioned area, precisely in the areas that would require stronger homeostatic adaptation (see Section 3.4). Thus, unraveling the spatial distribution of putative deficits in homeostatic plasticity is of particular relevance. Here, novel methods such as the measurement of functional E-I balance from electroencephalographic recordings [131] could be of direct relevance, indicating localized deficits that could then be corrected using neuromodulation. Alternatively, models such as ours could be used with patient-specific structural connectivity data and fitted to respective functional data by varying local parameters such as local target firing rates ($\rho$). Then, by comparing them with similar models with fully functioning homeostasis, regional differences could be detected, pointing to areas in need of further modulation of excitability. In any case, future studies should focus on measuring the evolution of E-I balance in the cortex of stroke patients, relating it to the recovery of function and evaluating possible impairments in homeostatic plasticity and how to correct them.

## 4.4. Emergence of biomarkers of stroke side-effects from E-I Homeostasis

Interestingly, we could relate certain side-effects of stroke and respective biomarkers with changes in the patterns of excitability observed in our model. Signatures such as increased excitability of the contralateral medial temporal cortex, the most common focus of epileptic seizures [104,108], could then be related to E-I homeostasis and to the tendency of stroke patients to developed seizures [3], in some cases evolving to epilepsy [107]. Critically, this finding is supported by one study in which neuromodulation was used in a rodent stroke model to increase motor cortex excitability [132]. While this led to a significant improvement in motor function, it also increased the propensity of the rodents to develop epileptic seizures. While this particular study was related to motor cortex excitability, its results are likely generalizable to other structures in the brain. Regarding depression, we observed a shift in the right-left asymmetry in motor cortex excitability towards higher excitability of the right side [112,113]. This was found particularly after right-side lesions in the frontal cortex, which are common in patients that experience post-stroke depression [110]. Interestingly, under the framework of E-I homeostasis, this result can be explained, since right lesions would lead to higher increases in excitability on the right side, thus leading to the observed changes in right-left asymmetry

associated with depression. Finally, chronic pain has been associated with maladaptive plasticity leading to a pathological increase in the excitability of sensorimotor cortices, thus creating a neuropathic sensation of pain [105]. In our case, we suggest that this process might not be maladaptive, but a physiological change that is required to compensate for a loss of cortico-cortical excitation, which could then affect how the sensorimotor areas respond to cortical and subcortical sensory input.

In addition, while the general effect observed was a widespread increase in excitability, our results show the surprising possibility that strong decreases in excitability can occur in certain regions for particular lesions. An example is decreased ipsilesional motor cortex excitability after a lesion in the precuneus or posterior cingulate cortex (S13 Fig). This particular case is interesting since chronic fatigue, commonly felt by stroke patients, has been associated with hypoexcitability of the motor cortex [11]. Therefore, we suggest that the participation of E-I homeostasis in enhancing post-stroke side effects may not only be tied to increased excitability, but also to the opposite effect in particular cases.

All that considered, care must be taken in attributing a causal relationship between the slow changes resulting from E-I homeostasis and the development of the mentioned side effects. Indeed, certain patients of stroke experience seizures already in the acute period, although this might be related to the excitotoxic release of glutamatergic neurotransmitters in this period [133]. Nonetheless, some patients continue experiencing repeated seizures into the chronic period [4], when such massive levels of glutamate are no longer present. In addition, the strongest risk factor of post-stroke depression is the amount of social support patients receive during recovery [109], seemingly rejecting changes caused by E-I homeostasis as a major cause for this pathology. Therefore, instead of attributing a fully causal role of E-I homeostasis in the emergence of the aforementioned side-effects, we suggest it as one of the multiple factors contributing to stroke symptomatology Alternatively, it is possible that the changes we observe could instead enhance the severity of said side-effects, caused by entirely different factors.

All that considered, we predict that E-I homeostasis, albeit necessary for post-stroke recovery, might inadvertently participate in the emergence of the discussed side effects. However, further research is required to understand this connection more clearly, for example, by associating particular lesions to specific patterns of alteration in excitability and the onset of the discussed pathologies in a patient-by-patient manner.

### 4.5. Limitations

One of the main caveats of our work is the discrepancy between the optimal value of mean conduction delay and the respective empirical values. More specifically, we used a mean delay of 4ms in our simulations, corresponding to a conduction speed of ≈39m/s. While such values can be found in axons of corticospinal tracts [134], the same may not apply to cortico-cortical axons. While the conduction velocity of long-range white matter tracts in the cortex is still unclear [134,135], recent results obtained by measuring the latency in evoked potentials after direct electrical stimulation estimate a mean delay of around 10 ms, corresponding to a median conduction velocity of 3.9 m/s [136]. Conversely, estimations based on axon diameters and g-ratio (inner to outer diameter of myelinated axons) of transcallosal axons suggest a median conduction velocity of 14 m/s, with some degree of heterogeneity between tracts [136,137]. While our obtained conduction velocity is in line with previous computational studies using coupled gamma oscillators [77], it is considerably faster than the empirical values found in literature. This is likely a result of the intrinsic frequency of oscillation of the neural masses (~40 Hz), which favors fast interactions, within the time of an oscillatory cycle of 25 ms. Accordingly, modelling studies based on coupled oscillators with intrinsic frequency in

the alpha range can support slower conduction velocities, as they are less disruptive of phase interactions between neural masses [70]. For this reason, interpretation of the conduction delays obtained in modelling studies such as ours should be done with care. In addition, results suggest a degree of heterogeneity in conduction velocities across the cortex [137,138], as opposed to the uniform values used in our study. Therefore, future modelling studies should consider the use of empirically-grounded conduction velocities that are heterogeneous across the cortical connectome.

Additionally, a relevant limitation in our study is the fact that we only simulate cortical gray matter lesions, by removing all the connections to and from a given cortical area. While this approach is common in lesion studies [27,68,88], it neglects the impact of white-matter disconnection. Indeed, a cortical lesion might not only affect the gray contained by its volume, but also white-matter tracts that pass through it and may connect other regions. Importantly, recent research suggests a greater relevance of white-matter disconnection in predicting future deficits, when compared to gray-matter loss [139]. However, in our case, without lesion-specific information about lesion volume and the white matter tracts it intercepts, it is not possible to estimate the extent of white matter damage. Therefore, future modeling studies should focus on the incorporation of realistic cortical lesions affecting both gray and white matter. In addition, regions in the AAL parcellation not only have different levels of connectivity, but also different volumes. Therefore, while lesions in, for example, the precuneus and the superior frontal cortex are both single-node in our simulations, in reality, the latter would involve a much larger volume. Nonetheless, while studies show that lesion volume has an impact on the extent of functional damage and subsequent recovery [140], it is arguable that the graph properties of lesioned areas have a significant influence as well [68,141]. Also, given the heterogeneity in the lesions applied in this study, we argue that it still retains validity in representing the wide range of post-stroke deficits and the participation of E-I homeostasis in recovery.

Another missing aspect in our study is the influence of sub-cortical dynamics. Studies have shown that the processes of diaschisis involve subcortical structures as well, such as the spread of thalamocortical dysrhythmia due to decreased excitation in thalamocortical networks [142]. In addition, subcortical lesions also have substantial effects on cortical dynamics [143], albeit not as strongly as cortical lesions. While studying such effects would be important, it is out of the scope of our study, given the difficulty in modeling subcortical structures at such a large-scale, due to their functional and structural heterogeneity. Recent approaches in embedding multiscale subcortical networks in mean-field models of the human cortex [144] might, however, prove useful to further study the effects of subcortical lesions and the participation of subcortical structures in post-stroke recovery.

A further caveat of our study is the reduced implementation of E-I homeostasis mechanisms focusing on inhibitory synaptic scaling. Our approach is common in large-scale modelling studies, not only in the implementation applied here [69,70], but also in the form of Feedback Inhibition Control [71,72], which functions under the same principle of adjusting local (or feedback) inhibition to regulate the firing rate of excitatory populations. In addition, further arguments in favor of our approach, are tied to the demonstrated importance of inhibitory homeostasis for cortical function [48,66] and the fact that a long-term decrease in inhibitory activity has been robustly observed in rodent stroke models [36] and patients [37–39]. More importantly, research suggests a correlation between the magnitude of this decrease and functional recovery [39]. Nonetheless, changes in excitatory neurotransmitters have been observed in stroke patients as well and different mechanisms of E-I homeostasis, such as excitatory synaptic scaling and regulation of intrinsic excitability [60] are likely involved. Further studies could then focus on the involvement of such mechanisms in stroke recovery, the magnitude of their participation, or the possibility that some of them, such as changes in intrinsic

excitability, come into play when other types of homeostasis are not sufficient to adapt to the damage. Moreso, literature indicates that significant structural rewiring is present in the post-stroke brain, from the peri-infarcted cortex to remote regions [145]. In addition, studies show the relevance of such structural plasticity mechanisms not only for behavioral recovery [146], but also for functional remapping [147] and the restoration of dynamics [28]. Therefore, future investigation could be devoted to the combined action and synergies of structural plasticity and E-I homeostasis in large-scale models of stroke.

Finally, an important caveat in the analysis is that we do not measure changes in homotopic interhemispheric connectivity, shown to be one of the strongest biomarkers of stroke correlated with patient behavior [21]. The main rationale behind this decision is the fact that large-scale computational models are generally lacking in the representation of interhemispheric homotopic connectivity in the cortex, likely due to an underestimation of white-matter tracts connecting the two hemispheres from methods such as diffusion tensor imaging [148]. Indeed, studies stress the importance of callosal white matter tracts in underlying stable homotopic FC and communication between hemispheres [149]. Therefore, to counteract the underestimation of homotopic white matter tracts, recent studies suggest the improvement of structural connectivity data with white-matter microstructure [150], the artificial augmentation of homotopic connections [151] or the use of effective connectivity [152], which corrects such differences in simulated data [153], and has shown to be a better predictor of post-stroke deficits [154]. Notwithstanding, we were able to replicate the effects of stroke [22] in FC graph properties relevant for cortical function, such as modularity or small-worldness [25,26], showing the participation of E-I homeostasis in their recovery. Therefore, despite the caveats, our biologically constrained model succeeds in answering stroke-specific physiological benchmarks leading to a new set of predictions that can be validated with current experimental methods.

## 5. Conclusion

In conclusion, our results lend strength to the claim that cortical E-I homeostasis is an important driver of stroke recovery, not only by showing that it corrects deficits in static properties of FC, but also that the required adjustments to local inhibition are consistent with the literature on post-stroke changes in inhibition. In addition, we suggest that specific patterns of altered excitability observed in our model can be associated with biomarkers of known side effects of stroke (e.g. seizures, depression, neuropathic pain), offering at least a partial explanation for the increased propensity of stroke patients to develop them. Therefore, by observing stroke through the lens of E-I homeostasis, we hope to advance the current knowledge about the neural processes involved in stroke recovery, essential to improve the effectiveness of therapeutical approaches that modulate cortical excitability, to predict more reliably the occurrence of stroke side effects and to better understand putative deficits in homeostatic plasticity that can hinder the rehabilitation process. Furthermore, we emphasize local E-I balance as a relevant variable for the study of emergence in cortical dynamics and stress the importance of employing a multiscale approach when exploring the governing principles underlying the emergent functional properties of the human brain in health and disease.

## Supporting information

**S1 Fig. Behavior of uncoupled Wilson Cowan node under different parameter combinations.** A) Impact of changing the parameter P which controls the intrinsic excitability of the Wilson Cowan node, on node activity and power spectrum. On the left side, we show results for models without noise and, on the right side, we show results of nodes with gaussian noise

with 0.01 standard deviation. Note that, in our model, uncoupled nodes go from a state of low activity to a limit cycle (by increasing P, showing the behavior of a Hopf bifurcation For the chosen population time constants $\tau_E$ = 2.5ms $\tau_I$ = 5.0ms the Wilson Cowan model displays oscillations at 40 Hz. B) Impact of changing population time constants on the oscillatory dynamics of uncoupled noisy Wilson Cowan nodes (Gaussian noise, 0.01 standard deviation) For all shown plots, $\tau_I = 2\tau_E$ It can be observed that the intrinsic frequency of oscillation of the Wilson Cowan nodes is changed by varying the time constants of the excitatory and inhibitory populations.
(TIF)

**S2 Fig. Change in local inhibitory weights caused by homeostatic plasticity for different time constants of homeostatic plasticity.** A) Variation in time in local inhibitory weights for all 78 nodes in the model, under different time constants of homeostatic plasticity, for the following combination of free parameters C = 4.07 ρ = 0.2 md = 4ms. Note that while $c_{EI}$ values take longer to reach a steady state for slower time constants, the final steady state values are virtually the same. B) Scatter plots of steady state c EI values for each homeostatic time constant against each other Note that values are virtually the same, showing that, as long as the homeostatic time constant is sufficiently slow to be decoupled from local node dynamics, it can be arbitrarily fast without affecting the steady state of the system.
(TIF)

**S3 Fig. Examples of the application of the steady-state detection algorithm for models with two different combinations of free parameters.**
(TIF)

**S4 Fig. Results of the application of different clustering algorithms to average functional connectivity from healthy subjects.** A) Resulting cluster inertia from applying the k-means algorithm described in the methods to empirical averaged functional connectivity from healthy subjects, with different numbers of clusters. Stars indicate potential 'elbows' in the cluster analysis, i.e. local minima or points with an inflection in inertia relative to the number of clusters. Inertia was calculated using the scikit learn module in Python. B) Resulting cluster distance from hierarchical clustering to averaged functional connectivity from healthy subjects, with different numbers of clusters. Stars indicate potential 'elbows' in the cluster analysis, i.e. local minima or points with an inflection in distance relative to the number of clusters. Hierarchical clustering was computed using the scikit learn module in Python. C) Dendrogram of averaged functional connectivity from healthy subjects. Colors represent 6 different clusters. D) Functional networks resulting from the application of the k means clustering algorithm to empirical data with 4 and 6 clusters. Note that the resulting networks for k = 6 can be equated to known resting state networks (e.g. visual (first) somatomotor (second) and default mode network (third)). E) Functional networks resulting from the application of hierarchical clustering to empirical data with 4 and 6 clusters. Note that the resulting networks for both k = 4 and k = 6 are reasonably similar to the ones in D), with known resting state networks emerging when k = 6.
(TIF)

**S5 Fig. Post-stroke change in modularity for different clustering algorithms, numbers of clusters and edge density threshold ranges.** A) Normalized modularity at T1 (acute post-lesion) and T2 (chronic post-lesion) for different results of k-means clustering. Each plot represents modularity analysis using as modules the result of k-means with the number of clusters ranging from 4 (left) to 10 (right). In each plot, we present results across a range of density thresholds and the average across density thresholds. Across density thresholds, asterisks

represent the level of significance of a Mann-Whitney U-test. For the average across density thresholds, asterisks represent the level of significance of a Wilcoxon ranked sum test against baseline (norm. mod. = 1). * $p<0.05$, ** $p<0.01$, *** $p<0.001$. B) Same as A), but for modules derived from hierarchical clustering. C) Normalized modularity at T1 (acute post-lesion) and T2 (chronic post-lesion) for edge-density thresholds ranging between 0.02 and 0.2, with 6 modules derived from k-means (Left) or hierarchical clustering (Right). In each plot, we present results across the range of density thresholds and the average across density thresholds. Across density thresholds, asterisks represent the level of significance of a Mann-Whitney U-test. For the average across density thresholds, asterisks represent the level of significance of a Wilcoxon ranked sum test against baseline (norm. mod. = 1). * $p<0.05$, ** $p<0.01$, *** $p<0.001$. D) Normalized modularity at T1 (acute post-lesion) and T2 (chronic post-lesion) using a weighted modularity algorithm. We present results across a range of density thresholds and the average across density thresholds. Across density thresholds, asterisks represent the level of significance of a Mann-Whitney U-test. For the average across density thresholds, asterisks represent the level of significance of a Wilcoxon ranked sum test against baseline (norm. mod. = 1). * $p<0.05$, ** $p<0.01$, *** $p<0.001$.
(TIF)

**S6 Fig. Results of fitting across the full parameter space.** Model fit over full parameter space. Each column of three plots represents the results of a grid search over the parameters of global coupling (*C*) and target firing rate (FR) ($\rho$), for a specific mean delay between 0 and 15 ms. In each column, model performance is shown according to the following metrics: (Top) Pearson's correlation between the upper triangle of simulated and empirical FC matrices, (Middle) mean squared error (MSE) between simulated and empirical FC matrices and (Bottom) Kolmogorov-Smirnoff (KS) distance between the distribution of values in simulated and empirical FCD matrices.
(TIF)

**S7 Fig. Model optimal points for individual-level fit.** The colormaps represent the model fit over full parameter space, obtained by comparing the average FC and aggregated FCD distribution from empirical data with model results. Model performance is evaluated according to the following metrics: (Left) Pearson's correlation between the upper triangle of simulated and empirical FC matrices, (Middle) mean squared error (MSE) between simulated and empirical FC matrices and (Right) Kolmogorov-Smirnoff (KS) distance between the distribution of values in simulated and empirical FCD matrices. The green dots represent the optimal point obtained from the same procedure when applied to subject-specific FC matrices and FCD distributions. Note the higher concentration of optimal points around the parameter region used as the optimal working point for our simulations (C = 4.07, $\rho$ = 0.2).
(TIF)

**S8 Fig. Functional connectivity recovery and reorganization.** Correlation between recovery and reorganization of FC across lesions. We classify differences between T1 and T2 as recovery, since they are related to the recovery of acute FC deficits and differences between T2 and T0 as reorganization, since they quantify the reorganization of FC connections that is required for recovery. From left to right, we show the correlation across lesions between SC and ΔFC (T2-T1), FC(0) and ΔFC(T2-T1), ΔFC(T2-T1) and ΔFC(T1-T0), SC and ΔFC(T2-T0), FC(T0) and ΔFC(T2-T0), FC(T0) and ΔFC(T2-T0) normalized to the maximum possible change, i.e. (FC(T2)-FC(T0))/(1 −FC(T0)). Dots represent the correlation between each of the combinations of measures on an individual lesion level. Dot color indicates the significance of the correlation, with green representing FDR corrected p-value < 0.05. Boxplots represent the

distribution across lesions. P-values above each boxplot represent the result of a Wilcoxon ranked-sum test, adjusted for multiple comparisons with FDR correction. Note that recovery is strongly negatively correlated with acute deficit in FC (third boxplot), indicating that deficits are being appropriately corrected from T1 to T2. In addition, reorganization is generally negatively correlated with healthy FC (T0) (fifth boxplot), meaning that reorganization is underlined by increases in the weakest healthy functional connections, which can be conceptualized as the formation of new functional connections. The significance of these results is maintained when normalizing for maximum possible increase in FC (sixth boxplot).
(TIF)

**S9 Fig. Correlation between structural graph properties of lesioned nodes and effects on functional connectivity.** A) Distance from baseline FC matrices at T1 (acute post-lesion) and T2 (chronic post-lesion) against node degree, betweenness centrality and clustering coefficient of lesioned nodes. All graph theoretical measures of lesioned nodes used in the plots were calculated using the *networkx* module in Python, after transforming the SC matrix into an undirected unweighted graph by thresholding the 10% strongest structural connections. B) Same as A), for the difference in correlation between structural and functional connectivity at T1 and T2, compared to baseline. C) Same as A), for normalized modularity at T1 (acute post-lesion) and T2 (chronic post-lesion). Normalization was calculated using the value at T0 as the baseline.
(TIF)

**S10 Fig. Changes in excitability of middle temporal cortex across lesions.** Variation, between T0 and T2, in $c_{EI}$ weight of the middle temporal cortex after lesion in the same hemisphere. Points represent results for left and right lesions in the respective areas and the dashed line represents the average between these two values. Areas are ordered according to the average effect on middle temporal cortex excitability.
(TIF)

**S11 Fig. Change in asymmetry of motor cortex excitability across lesions.** Variation, between T0 and T2, in motor cortex (precentral gyrus) excitability asymmetry across all lesions. Positive values indicate that the left motor cortex experienced a stronger increase in excitability when compared to its right counterpart, while negative values indicate the opposite variation. Areas are ordered according to lesion effects in this asymmetry.
(TIF)

**S12 Fig. Change in excitability of somatosensory cortex across lesions.** Variation, between T0 and T2, in $c_{EI}$ weight of the somatosensory cortex (postcentral gyrus) after lesion in the same hemisphere. Points represent results for left and right lesions in the respective areas and the dashed line represents the average between these two values. Areas are ordered according to the average effect on somatosensory cortex excitability.
(TIF)

**S13 Fig. Changes in excitability of ipsilesional motor cortex across lesions.** Variation, between T0 and T2, in $c_{EI}$ weight of the motor cortex (precentral gyrus) after lesion in the same hemisphere. Points represent results for left and right lesions in the respective areas and the dashed line represents the average between these two values. Areas are ordered according to the average effect on motor cortex excitability.
(TIF)

**S14 Fig. Post-lesion adaptation time is dependent on lesion strength.** A) Example of the evolution of local inhibitory weights ($c_{EI}$) after lesion in a high strength node (Superior frontal

gyrus, medial orbital) and a low strength node (Left Olfactory Cortex). B) Correlation between lesion strength and time required for $c_{EI}$ stabilization after lesion. Lesions in nodes with strong structural connections require longer adaptation times for the restoration of local E-I balance. (TIF)

**S1 Text. Description of test condition for detection of steady states in $C_{EI}$.**
(PDF)

## Author Contributions

**Conceptualization:** Francisco Páscoa dos Santos, Paul F. M. J. Verschure.

**Formal analysis:** Francisco Páscoa dos Santos, Jakub Vohryzek.

**Funding acquisition:** Paul F. M. J. Verschure.

**Methodology:** Francisco Páscoa dos Santos, Jakub Vohryzek.

**Software:** Francisco Páscoa dos Santos.

**Supervision:** Paul F. M. J. Verschure.

**Visualization:** Francisco Páscoa dos Santos.

**Writing – original draft:** Francisco Páscoa dos Santos.

**Writing – review & editing:** Francisco Páscoa dos Santos, Jakub Vohryzek, Paul F. M. J. Verschure.

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
