## [Decision Letter · Decision Letter 0]

9 Jan 2023

Dear Dr. dos Santos,

Thank you very much for submitting your manuscript "Multiscale Effects of Excitatory-Inhibitory Homeostasis in Lesioned Cortical Networks: A Computational Study" for consideration at PLOS Computational Biology.

As with all papers reviewed by the journal, your manuscript was reviewed by members of the editorial board and by several independent reviewers. In light of the reviews (below this email), we would like to invite the resubmission of a significantly-revised version that takes into account the reviewers' comments.

Reviewers all agree that the work is interesting and beneficial, however, there are several aspects that the authors should address as outlined below:

We cannot make any decision about publication until we have seen the revised manuscript and your response to the reviewers' comments. Your revised manuscript is also likely to be sent to reviewers for further evaluation.

Sincerely,

Hayriye Cagnan

Academic Editor

PLOS Computational Biology

Lyle Graham

Section Editor

PLOS Computational Biology

Reviewers all agree that the work is interesting and beneficial, however, there are several aspects that the authors should address as outlined below:

Reviewer's Responses to Questions

**Comments to the Authors:**

Reviewer #1: This study combines elegantly many different analyses methods from the fields of network neuroscience and computational neuroscience. I congratulate the authors on this fine well-written work. I find this modeling study interesting with dense results that identify a potential mechanism to explain (some parts of) post-stroke recovery. Find some major and minor comments below.

a) Is there post lesion an increase in excitation or loss in excitation? I find contradictory statements in the first and last sentence of the second paragraph on the second page (lines 71-82).

b) Why did the authors choose AAL parcellation? It seems like an outdated choice.

c) Why not take the same subjects from the HCP database for the structural and functional matrices?

d) And why not perform even an individual-level fitting here? Since one can assume that the SC-FC coupling should in the individual case be even higher.

e) Where did the authors explain how they obtain the tract lengths?

f) Please explain all variables in Eq. (1): C_EI,i what is that? What is N? what is P? what is i?

g) What is the F-I curve?

h) The described mechanism of E-I homeostasis sounds similar to the mechanism of feedback inhibition control, which is used often to control for over-excitation in modelling studies. Could the authors please comment on the difference between their proposed mechanism and FIC? (e.g. Deco et al, 2014, Deco, G., Ponce-Alvarez, A., Hagmann, P., Romani, G. L., Mantini, D., & Corbetta, M. (2014). How local excitation–inhibition ratio impacts the whole brain dynamics. Journal of Neuroscience, 34(23), 7886-7898.)

i) Why did the authors choose logarithmically spaced values for C?

j) Line 270: typo matrices

k) Please explain abbreviations in Fig. 1 in the legend.

l) Another modelling aspect the authors could try to relate to empirical stroke patient finding would be the lengths of the stabilization period in their different virtual lesion simulations. If the authors would have any interpretation about what time frame we are looking at in the simulations and compare the relative stabilization period lengths with literature about which regional lesions are known to have a faster recovery relative to other regions. This could be an interesting additional aspect that the authors could check.

m) Though the authors mention “multiscale” in the title, they do not use it in the main text except for pointing towards another study. I suggest to remove it from the title or explain more directly in the manuscript how the word is to be understood in this study.

Reviewer #2: Authors explore the effects of excitatory-inhibitory homeostasis in a whole-brain model based on the connectome and the WC model. The results are not directly linked to empirical data, but the computational exploration is nevertheless technically very sound and solid and the paper is very well written.

The framing of the problem, and the link with the empirical studies is solid.

However, I still have some issues that I would like to see being resolved before I suggest a publication of the paper.

- I don’t see a justification for using the adjective multiscale in the title.. even though the mechanisms of the E/I homeostasis are on microscale, still the WC model is quite phenomenological in order to call the mechanisms multiscale.

- why the graph theory metrics are applied on thresholded and binarized matrices, when the same metrics are available for weighted matrices in the brain connectivity toolbox for example?

- the optimal conduction speed of 39m/s is unreasonably too high for cortico-cortical axons… for example, it is almost 10 times higher than the value obtained from the largest database of cortico-cortical evoked potential (Le Marechal et al Brain 2021), and the same mean value of ~3.3m/s for example was shown to be needed to reproduce some the major frequency-dependent activation patterns in Petkoski et al Netw Neurosci 2022. I’m aware that some other simplifications (e.g. setting 40Hz in every region) in the model might be responsible for such a high value, but I expect the issue of the propagation velocity to be at least properly discussed..

- l499 is there any indication that it is more new connections, or strengthening of the old one? It should be checked in the model, but also linked to what is known in the literature.. Similar to this it should be discussed that structural rewiring is another phenomena that might shape the changes in FC due to stroke.. for example it is known that there is a contralateral increase of connectivity in stroke (e.g. see the review by Nudo et al 2013), and this was then used to model FC changes of calcium activity in stroke in mice in Allegra et al Front Sys Neurosci 2020.. also in the same study the changes of neuromodulation were used as mechanism for modelling the pericital local changes in the dynamics..

- it is worth at least mentioning that effective connectivity has been shown to be better predictive for stroke than FC, Adhikari et al Brain 2021.

- I applaud the authors for their focus on FCD, even though as they admit there is still not much empirical results for this case.. maybe they should also mention that FCD is becoming proven as more informative than FC for ageing, cognitive performance, etc.. However, they seem to have omitted a very recent work that uses FCD based metric, Kim et al Nat Comm 2022.

- when discussing Fig. 5 A, authors should be clearer that the Euclidian distance is relevant only because the connectivity decreases with the distance, and the topology is the only determinant of their results.

- when discussing the importance of the graph theory metrics and topology for the stroke impact, it is maybe interesting to make a link that the same is the case for number of lesions to control seizure propagation in epilepsy, which is highly correlated with the graph theory properties of the node, Olmi et al Plos CB 2019.

- as for the limitations, it is worth mentioning the fact that they assume same strength of homeostasis at every brain region, while as I mentioned in the previous comments, it is expected regions strongly connected to the stroke region, and especially the same contralateral region, to be more affected by structural alterations, which don’t exclude the E/I homeostasis.

- l681 using further as a verb doesn’t sound right.

**Have the authors made all data and (if applicable) computational code underlying the findings in their manuscript fully available?**

Reviewer #1: Yes

Reviewer #2: Yes

PLOS authors have the option to publish the peer review history of their article (what does this mean?). If published, this will include your full peer review and any attached files.

Reviewer #1: **Yes: **Jil Mona Meier

Reviewer #2: No
---

## [Decision Letter · Decision Letter 1]

5 Jun 2023

Dear Dr Pascoa dos Santos,

Thank you very much for submitting your manuscript "Multiscale Effects of Excitatory-Inhibitory Homeostasis in Lesioned Cortical Networks: A Computational Study" for consideration at PLOS Computational Biology. As with all papers reviewed by the journal, your manuscript was reviewed by members of the editorial board and by several independent reviewers. The reviewers appreciated the attention to an important topic. Based on the reviews, we are likely to accept this manuscript for publication, providing that you modify the manuscript according to the review recommendations.

While both reviewers are happy with the revisions, could you please expand your discussion on conduction velocity?

Sincerely,

Hayriye Cagnan

Academic Editor

PLOS Computational Biology

Lyle Graham

Section Editor

PLOS Computational Biology

While both reviewers are happy with the revisions, could you please expand your discussion on conduction velocity?

Reviewer's Responses to Questions

**Comments to the Authors:**

Reviewer #1: The authors have addressed my comments to my satisfaction. I am impressed with the addition of new results based on my comments and thank the authors for their additional work, which in my opinion strengthened the manuscript.

Reviewer #2: The authors have seriously taken all of my comments, and they have in-depth addressed all of them.

The only remaining issue is remaining the conduction velocity. Most of the cited references by the authors in their response, do not correspond to cortico-cortical neurons, but also to the spinal cord, where those high velocities are recorded. see for example http://www.scholarpedia.org/article/Axonal_conduction_delays

or as I mentioned, I still consider the Lemarcehal et al. Brain paper as an actual gold standard, since it does measure the velocity in human over the whole cortex. This time I’m also sending the link for the paper and it’s doi

https://pubmed.ncbi.nlm.nih.gov/35416942/

doi: 10.1093/brain/awab362

Referring to purely computational studies, where the velocity is just fitted to get a good correspondence (as in the mentioned Cabral paper) is a weak argument, though I do understand that due to different approximation sometimes models might require non-realistic values for the conduction velocity. And that is still acceptable (for me at least) for as long as it is properly discussed.

**Have the authors made all data and (if applicable) computational code underlying the findings in their manuscript fully available?**

Reviewer #1: Yes

Reviewer #2: None

PLOS authors have the option to publish the peer review history of their article (what does this mean?). If published, this will include your full peer review and any attached files.

Reviewer #1: **Yes: **Jil Meier

Reviewer #2: No

Figure Files:

Data Requirements:

Reproducibility:

References:

---

## [Editor Report · Decision Letter 2]

18 Jun 2023

Dear Dr Francisco Pascoa dos Santos,

We are pleased to inform you that your manuscript 'Multiscale Effects of Excitatory-Inhibitory Homeostasis in Lesioned Cortical Networks: A Computational Study' has been provisionally accepted for publication in PLOS Computational Biology.

Best regards,

Hayriye Cagnan

Academic Editor

PLOS Computational Biology

Lyle Graham

Section Editor

PLOS Computational Biology

---

## [Editor Report · Acceptance letter]

30 Jun 2023

PCOMPBIOL-D-22-01745R2 

Multiscale Effects of Excitatory-Inhibitory Homeostasis in Lesioned Cortical Networks: A Computational Study

Dear Dr Páscoa dos Santos,

I am pleased to inform you that your manuscript has been formally accepted for publication in PLOS Computational Biology. Your manuscript is now with our production department and you will be notified of the publication date in due course.

With kind regards,

Zsofi Zombor
